METHODS AND RESOURCES

# Low-cost, versatile, and highly reproducible microfabrication pipeline to generate 3D-printed customised cell culture devices with complex designs

Cathleen Hagemann[1,2,3°], Matthew C. D. Bailey[2,4°], Eugenia Carraro[1,2,3], Ksenia S. Stankevich[5], Valentina Maria Lionello[2,6], Noreen Khokhar[2,6,7], Pacharaporn Suklai[1,2,3], Carmen Moreno-Gonzalez[1,2,3], Kelly O'Toole[1,2,3], George Konstantinou[2], Christina L. Dix[2], Sudeep Joshi[1,2], Eleonora Giagnorio[2,6,8], Mads S. Bergholt[4], Christopher D. Spicer[5], Albane Imbert[2], Francesco Saverio Tedesco[2,6,9], Andrea Serio[1,2,3]*

1 United Kingdom Dementia Research Institute Centre, Institute of Psychiatry, Psychology and Neuroscience, King's College London, Maurice Wohl Clinical Neuroscience Institute, London, United Kingdom, 2 The Francis Crick Institute, London, United Kingdom, 3 Dementia Research Institute (UK DRI), 4 Centre for Craniofacial & Regenerative Biology, King's College London, London, United Kingdom, 5 Department of Chemistry and York Biomedical Research Institute, University of York, York, United Kingdom, 6 Department of Cell and Developmental Biology, University College London, London, United Kingdom, 7 Randall Centre for Cell and Molecular Biophysics, King's College London, London, United Kingdom, 8 Neurology IV—Neuroimmunology and Neuromuscular Diseases Unit, Fondazione IRCCS Istituto Neurologico Carlo Besta, Milan, Italy, 9 Dubowitz Neuromuscular Centre, UCL Great Ormond Street Institute of Child Health & Great Ormond Street Hospital for Children, London, United Kingdom

☯ These authors contributed equally to this work.
* andrea.serio@kcl.ac.uk

**Data Availability Statement:** All data and procedures are available within the paper and additional information and designs are available as

## Abstract

Cell culture devices, such as microwells and microfluidic chips, are designed to increase the complexity of cell-based models while retaining control over culture conditions and have become indispensable platforms for biological systems modelling. From microtopography, microwells, plating devices, and microfluidic systems to larger constructs such as live imaging chamber slides, a wide variety of culture devices with different geometries have become indispensable in biology laboratories. However, while their application in biological projects is increasing exponentially, due to a combination of the techniques, equipment and tools required for their manufacture, and the expertise necessary, biological and biomedical labs tend more often to rely on already made devices. Indeed, commercially developed devices are available for a variety of applications but are often costly and, importantly, lack the potential for customisation by each individual lab. The last point is quite crucial, as often experiments in wet labs are adapted to whichever design is already available rather than designing and fabricating custom systems that perfectly fit the biological question. This combination of factors still restricts widespread application of microfabricated custom devices in most biological wet labs. Capitalising on recent advances in bioengineering and microfabrication aimed at solving these issues, and taking advantage of low-cost, high-resolution desktop resin 3D printers combined with PDMS soft lithography, we have developed an optimised a low-cost and highly reproducible microfabrication pipeline. This is thought

part of this publication and associated files, as well as available to download from the corresponding GitHub repository (https://github.com/SerioLab/SOL3D).

**Funding:** The Serio lab acknowledge support of the UK Biotechnology and Biological Sciences Research Council (BBSRC) [BB/T014318/1] [BB/W006561/1] and of the Dementia Research Institute (UKDRI). F.S.T and A.S. are part of the Horizon Europe "MAGIC" consortium (101080690; www.magic-horizon.eu); this work is funded by UK Research and Innovation (UKRI) under the UK government's Horizon Europe funding guarantee grant numbers 10080927, 10079726, 10082354 and 10078461. Work in the Tedesco lab was also supported by the European Research Council (759108), AFM-Telethon (21687), BBSRC (BB/M009513/1), CureCMD (576031), Muscular Dystrophy UK and the NIHR (the views expressed are those of the authors and not necessarily those of the National Health Service, the NIHR, or the Department of Health). This research was funded in whole, or in part, by the Wellcome Trust. We would like to acknowledge the Making Lab facility, a Science Technology Platform at the Francis Crick Institute; A.S, A.I and F.S.T acknowledge support by the Francis Crick Institute, which receives its core funding from Cancer Research UK, the UK Medical Research Council (MRC) and the Wellcome Trust (CC0102). K.S.S and C.D.S acknowledge funding from the Leverhulme Trust (RPG-2022-174). C.D.S acknowledges generous support through a Wellcome Trust Career Development Award (225257/Z/22/Z). The funders had no role in study design, data collection and analysis, decision to publish, or preparation of the manuscript.

**Competing interests:** The authors have declared that no competing interests exist.

**Abbreviations:** EB, embryoid body; ECM, extra cellular matrix; GS, goat serum; ICC, immunocytochemistry; IPA, isopropanol; iPSC, induced pluripotent stem cell; MN, motor neuron; MNP, motor neuron progenitor; PDMS, polydimethylsiloxane; SOL3D, softlithography on 3D vat polymerised moulds.

specifically for biomedical and biological wet labs with not prior experience in the field, which will enable them to generate a wide variety of customisable devices for cell culture and tissue engineering in an easy, fast reproducible way for a fraction of the cost of conventional microfabrication or commercial alternatives. This protocol is designed specifically to be a resource for biological labs with limited expertise in those techniques and enables the manufacture of complex devices across the μm to cm scale. We provide a ready-to-go pipeline for the efficient treatment of resin-based 3D-printed constructs for PDMS curing, using a combination of polymerisation steps, washes, and surface treatments. Together with the extensive characterisation of the fabrication pipeline, we show the utilisation of this system to a variety of applications and use cases relevant to biological experiments, ranging from micro topographies for cell alignments to complex multipart hydrogel culturing systems. This methodology can be easily adopted by any wet lab, irrespective of prior expertise or resource availability and will enable the wide adoption of tailored microfabricated devices across many fields of biology.

## 1. Introduction

Stem cell-based models are an invaluable resource, which allow the study of nearly any cell type in vitro [1–4]. The advent of cellular reprogramming and subsequent access to patient-derived stem cell models have also galvanised their position as an ideal tool to investigate cellular processes in health and disease [5–9]. While stem cell models offer significant control over the identity of cultured cell types, the conventional culture systems used for them typically lack the ability to control key parameters of the culture itself, which greatly influence the analysed biological processes. These parameters include the relative position of the cultured cells, grouping, cell–cell and cell–material interactions and many others, depending on the biological questions asked. Although a plethora of commercially devices allow some degree of control over culture conditions, they are often non-customisable and require adaptation of the biological experimental parameters to the specific device characteristics, rather than the more desirable opposite.

Several bioengineering and fabrication strategies have been developed to create custom-engineered culture environments that direct the cell's environment [10,11] and they have enabled countless new biological insights. Most of these strategies include 2 key parts: a suitable material than can be made biocompatible and a method for shaping it into the desired forms. PDMS (polydimethylsiloxane) is a biocompatible, optically clear silicon-based elastomer with tuneable stiffness (800 kPa—10 MPa), compatible with multiple chemical modifications suitable for cell culture, and represents the most widely used material for fabrication of microfluidic devices and countless other culture platforms for investigating complex cellular interactions in vitro, while the biocompatibility also depends on the used cell type and surface modification [12,13]. Most of these cell culture devices and platforms have features ranging from micrometres to several millimetres, depending on the size of the cells, the required volume of the culture medium, and experimental paradigm. Being able to fabricate custom devices with both micron, millimetre and in some case centimetre scale opens countless possibilities for biological experiments but this often requires the combination of multiple techniques, and in some case, a variety of different expertise and equipment or resources that are not common in biologically focused labs. Consequently, it remains challenging for wet labs to

perform rapid prototyping of user-handleable macroscale devices with microscale features for cell culture applications.

Capitalising on recent important advancements in bioengineering and microfabrication aimed at solving these issues, and taking advantage of low-cost, high-resolution methods, we have developed an optimised a low-cost and highly reproducible microfabrication pipeline, thought specifically for biomedical and biological wet labs with not prior experience in the field, which will enable them to generate a wide variety of customisable devices for cell culture and tissue engineering with features that vary from 20/50 μm to several centimetres using 3D vat polymerisation.

We present here a detailed guide on how to use this pipeline, intended for biological and biomedical wet labs, together with the necessary information and context on the techniques involved across the fields of bioengineering and microfabrication and several application examples. What follows is a short contextualisation of the relevant technologies, in abridged form, and we provide an in-depth explanation and description of the available systems within the Supplementary guide (**S1 Text**).

In short, conventional microfabrication generates micron and millimetre-scaled features with a combination of photolithography and soft lithography methods. Particularly photolithography requires specialised facilities and expertise, and generally allows to only generate features within 1 scale at the time, as it is based on serial deposition of photo sensible polymers of defined thickness ranging from 1 μm to generally 500 microns maximum. While photolithography is essential for creating many popular cell culture devices, the specialised equipment, relatively long timescales, and expertise needed, limits wider adoption of custom microdevices [11].

3D printing technology has emerged as an accessible and adaptable tool for fast prototyping and fabrication of small objects. Alongside their increasing availability, rapid technological advancements in 3D printers have led to the development of several open-source projects aiming to enable any wet lab to create and adopt critical and innovative modelling strategies [14–19]. This is particularly important when considering the challenges experienced by laboratories in less developed countries in sourcing equipment or specific consumables, or the sometimes-steep practical barrier that some labs encounter when venturing into cell culture and biology from a different field.

There are a variety of different 3D printing techniques, ranging from filament deposition to vat polymerisation of resins. Most of these techniques are commercially available but more complex and intricate methods have been developed over time, such as two-photon-based microfabrication [20–23]. Although these custom-built, high-resolution setups are significant technical advancements, we focus here on vat polymerisation as it represents the most accessible and cost-effective form of 3D printing with a μm-scale resolution and provide further details on the other techniques as a complete overview in Supplementary guide 1 (**S1 Text**). Vat polymerisation is using a layer-by-layer construction of complex volumes using a UV curable resin as a material. Features are built though shining UV light on a thin volume of resin with a build plate next to it. The resin cures and attaches to the build plate. The plate then moves and gives space for a new layer of resin that is polymerised on to the layer of resin from before. The final construct is printed by using patterned UV light that resembles the end design layer by layer.

Unfortunately, most commercially available resins for UV vat polymerisation are cytotoxic and cannot be used for cell culture applications [24]. Additionally, the composition of these resins is often proprietary, and conversion or production of biocompatible resins requires skills limited to dedicated chemistry laboratories [25]. Some biocompatible resins are commercially available; however, they tend to be sold at a much higher cost than even high-resolution

resin, and more than the actual printers in some cases (e.g., Phrozen sonic mini 4K printer = GBP 365 [26], 1L Zortrax Raydent Crown and Bridge resin = GBP 392 [27]—prices at time of writing, for illustration only), undermining the applicability of 3D SLA printing for cell culture purposes (**S1 Fig**). Moreover, unlike PDMS and other silicon-based materials used for soft-lithography resins do not have tuneable stiffness and are generally not optically clear, factors that prevent their ability to act as a suitable substrate for cell culture or microscopy.

One possible solution to these problems would be to combine PDMS with UV resin vat polymerised 3D-printed moulds and effectively employ 3D printing in lieu of photolithography in conventional pipelines. However, curing of PDMS on vat polymerisation resin prints can be challenging as constituents of most commercially available resins inhibit PDMS polymerisation [28–31]. Furthermore, this curing inhibition makes demoulding difficult and can result in leaching of cytotoxic uncured PDMS monomers into the cell culture medium of even successfully demoulded designs [32], making the devices unusable for cell culture applications.

To overcome these challenges and facilitate the production of complex 3D constructs suitable for cell culture, several successful post processing and coating approaches have been attempted [33–35]. However, these protocols generally involve either long heat and detergent treatments [36], which often cause print deformation, or expensive techniques [37], not accessible to every lab. Others have circumvented this issue by using custom-made resins [38]. One example of the latter, coating of UV resin vat polymerised prints with parylene is effective in creating usable moulds and is sufficient to overcome curing inhibition of PDMS [39], but requires the use of specialised equipment and adds another potentially technically challenging step to optimise.

Driven initially by our own experience with adapting microfabrication techniques to hard biological questions and inspired by the many recent technical advancements by different groups, we aimed to create an optimised and universally effective pipelines that would include the production and post processing protocol for **so**ft **l**ithography on **3D** vat polymerised moulds (**SOL3D**), using a low-cost commercially available printer and materials.

To demonstrate the applicability of this method to several different biological experiments and provide an effective ready-to-use pipeline for other labs that do not have expertise in microfabrication, we demonstrated its use to develop customisable culture devices ranging from μm to mm and cm scale, with complex 3D shapes or and micro topographies. Together with the detailed protocols, we also provide the designs for each device showcased, which can be customised to fit different experimental needs.

## 2. Results

### 2.1 Optimisation of PDMS curing on 3D SLA printed moulds

To overcome the current barriers preventing the integration of 3D vat polymerisation for the fabrication of tissue culture constructs in biology labs, we aimed to optimise an easy-to-implement and widely applicable protocol to enable efficient PDMS curing on vat polymerised moulds using commercially available equipment. We, therefore, tested a variety of commercially available resins (Table 1) subsampling different manufacturers together with a commercially available high-resolution 3D vat printer (300 to 400 GBP retail price, Phrozen 4K Sonic Mini or Anycubic Photon S equivalent to approx. 2 batches of a monoclonal antibody for immunostaining).

First, we verified the previously reported cytotoxicity of each resin following conventional post processing steps (isopropanol washing, UV curing), either in an untreated state or with supplementary heat treatment, washing and UV sterilisation, by co-culturing chips of resin with induced pluripotent stem cells (iPSC)-derived motor neurons (MNs) (**S2 Fig**).

**Table 1. Resins.**

| Supplier | Resin | Synonym |
|---|---|---|
| Phrozen | Aqua Gray 4K | A |
| Elegoo | ABS-like | B |
| Anycubic | Clear | C |
| Liqcreate | Flexible X | D |
| Liqcreate | Premium Tough | E |
| Phrozen | Water-washable model Gray | F |
| NextDent | Ortho Clear | |

Additionally, we tested a grade 2a biocompatible resin used for dental implants. After 4 days of coculture with this resin, the iPSC-derived MNs show despite different washing protocols toxic effects, and a clear visual increase in debris compared to the control well (S3 Fig). As none of the resins were suitable for cell culture applications in direct use, we then focused on optimising the post-print processing protocol for resin moulds testing different parameters across 3 main steps: resin washing, print coating, and PDMS heat treatment curing (Fig 1A), to find an easy and fast method overcoming PDMS curing inhibition, as no standardised post processing protocol exists (Fig 1B). Resins were printed using modified manufacturer's settings for the recommended printer (either Anycubic or Phrozen) (Fig 1C and Table 1). We tested PDMS curing on the moulds at 6 different time points (2 h, 4 h, 6 h, 18 h, 22 h, 24 h) and considered the sample conditions not optimal for curing if the process took over 30 h.

The isopropanol washing step is designed to remove excess uncured resin from the printed moulds. We tested 2 different methods for removal, sonication and stirring, alone or in combination, each for 10 min, and found that post-printing washing conditions in isolation have a modest effect on PDMS curing time, but a combinatorial treatment was beneficial. Therefore, all subsequent experiments were performed using sequential treatment with sonication and washing (10 min each). Interestingly, we found that resin selection had more impact on curing time than washing itself, with resins A and F performing the best (Fig 1D). Washing of resin E and D was unsuccessful in most conditions, due to the amount of uncured resin adhering to the print from improper printing, and subsequent analysis of PDMS curing on these samples would bias the curing time if curing can take place at all.

It has been suggested that curing inhibition on resin can result from vaporised acrylate monomers [33], which are components of most resins, released into the PDMS during heating. We reasoned that either blocking the contact sites between acrylates and PDMS or reducing the release of acrylates from the resin during curing could be sufficient to allow efficient curing of PDMS on the moulds. To test these hypotheses, we used commercially available enamel paint to homogenously coat the washed 3D prints with an airbrushing system, forming a protective barrier between the PDMS and resin. We then compared the PDMS curing time of coated prints to uncoated prints at 3 different temperatures (60, 75, and 90°C), which allowed us to identify the role of temperature on acrylate release and PDMS curing. These experiments showed that enamel paint coating enabled PDMS curing not only on the surface but also throughout the whole cast and decreased PDMS curing times for all resins. It is important to note that Resin A is a special case, as it showed good PDMS curing performance with and without coating (Fig 1E), permitting the use of our post processing protocol with and without enamel paint. These 2 protocols differ not only in the coating but also in the curing temperature used, which impacts the overall manufacturing times. The benefit of the missing paint layer of the non-coating protocol is that it allows the manufacture of detailed PDMS moulds (features <300 μm) (Fig 1G). Overall, lower temperatures improved PDMS curing times on

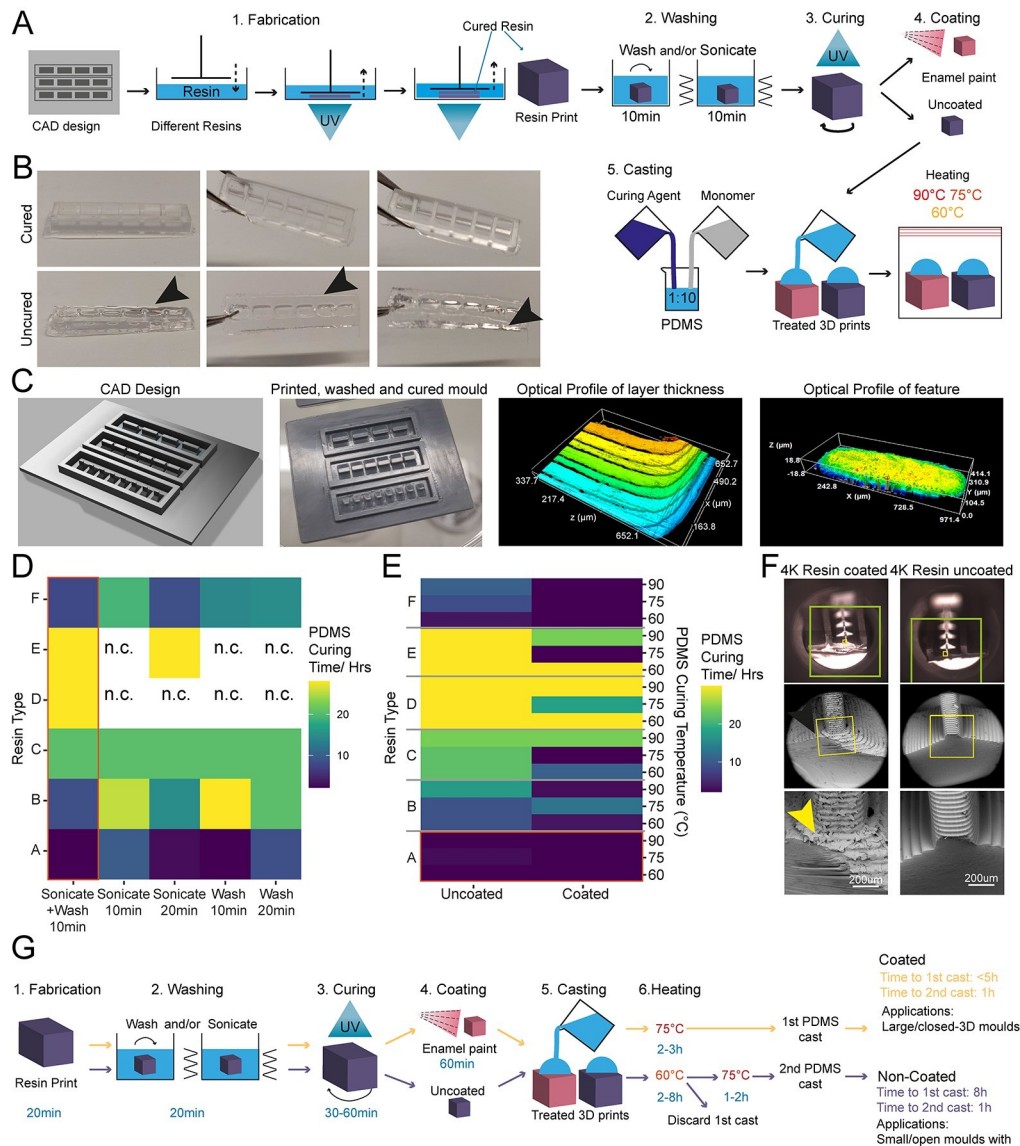

**Fig 1. Enamel paint coating facilitates rapid PDMS curing on 3D-printed moulds.** (A) Schematic overview of the investigation strategy to establish a protocol for PDMS curing on 3D-printed moulds. (B) Representative images of PDMS casts removed from printed devices classified as cured or uncured. Arrows highlight liquid PDMS. (C) Representative images of a CAD of 3D-printed moulds, the completed print, and surface optical profiles of layer thickness and feature dimensions. (D) Heatmap of PDMS curing time by resin type for different washing conditions. (E) Heatmap of PDMS curing time by resin type for different PDMS curing temperatures (right y-axis) and different SLA print coatings. (F) Representative SEM images of uncoated (right) and enamel paint coated prints (left), arrows highlight the paint layer. (G) Schematic overview of optimised fabrication, post processing, and PDMS casting protocols with (yellow) and without (purple) enamel coating.

resins and the variation in the effect of temperature on coated samples was negligible. Additionally, we observed that high temperatures (90°C) resulted in a significant warping of the print. These unwanted effects were less prominent at lower temperatures (60 to 75°C), which were still effective enough to cure PDMS.

It has also been reported that acrylate monomers and photo-initiators, which are resin components, can leach from the resins into PDMS, impacting the biocompatibility of cast

constructs [32]. To verify that the cured PDMS samples did not have leachates of resins or enamel paint, we used Raman spectroscopy to characterise the chemical composition of samples from each post processing condition and compared them to cured and uncured PDMS (S4 Fig). This spectral analysis revealed no detectable carryover of resin constituents or paint into the casted PDMS and high similarity of resin casts to cured PDMS. Measuring leachates from PDMS samples swelled in hexane for extended periods of time further demonstrated that leaching was minimal, with results suggesting that curing was >97% complete (see M&M). Similar experiments incubating PDMS samples cured in our resin moulds in aqueous media at 37°C for 72 h led to undetectable levels of leachates as assessed by MALDI-TOF mass spectrometry analysis (S5 Fig), while culturing resin print cast PDMS substrates with iPSC-derived MNs showed good biocompatibility over longer culturing periods (S6 Fig).

As introducing a layer of enamel coating on the mould devices could negatively impact fine feature sizes, we used SEM imaging on printed moulds with resin A to quantify whether print dimensions and surface roughness were significantly affected. Analysis showed that coated prints exhibited a thin layer of paint (between 30 and 50 μm) and greater surface roughness compared to uncoated ones (S7 Fig). This limits the application of this method to features larger than 100 μm in any dimension (Fig 1F).

Overall, we established a fast and robust post processing pipeline, identified suitable resins and provide a ready-to-use protocol for 3D vat printing. We provide 2 modified versions of an effective post processing protocol for **so**ft-**l**ithography on **3D** vat polymerised moulds, depending on the design and feature size (Fig 1G).

## SOL3D fabrication allows the generation of complex 3D-shaped stencils for precise control of cell positioning and grouping within open wells

**3D-printed stencil-aided dry plating devices to control cell location and number within standard well plates.** Conventional open well culture systems generally do not allow control over cell position, grouping, and numbers in an easy and reproducible fashion, limiting the complexity of in vitro modelling experiments. Several techniques are available to overcome these limitations and to create precise arrangements of cells within culture vessels—from microfluidic devices to cell bioprinting—; however, most rely on creating compartmentalised structures that limit the manipulation of the cells granted by the open well systems.

A different approach that allows both to increase complexity within conventional culture vessels and to maintain an open well system are stencil-like plating devices [40,41]. These systems are temporary structures that guide the organisation of cells within an open well, although at present they suffer from the same fabrication limitations as the abovementioned strategies. Moreover, this method is especially affected by the technical limitations of feature sizes and aspect ratios dictated by photolithography, resulting in thin devices that are difficult to handle and have limited customisation possibilities (S8 Fig).

We decided to model these types of devices, using human-induced pluripotent stem cell-derived MNs as a cell model system, for an initial proof-of-principle of our optimised SOL3D protocol, based on an engineered platform we recently developed for MN cultures using a micropatterned substrate to facilitate axonal elongation [42].

We combined and optimised this platform with our SOL3D moulding protocol to create a tailored plating strategy for investigating hiPSC-derived MN behaviour with control over cell location and orientation. We designed moulds for casting PDMS stencil-well devices, rectangular extruded features with funnel-shaped media reservoirs as complex 3D features to ease cell seeding. This optimised design permits rapid and facile manual seeding as cells can settle

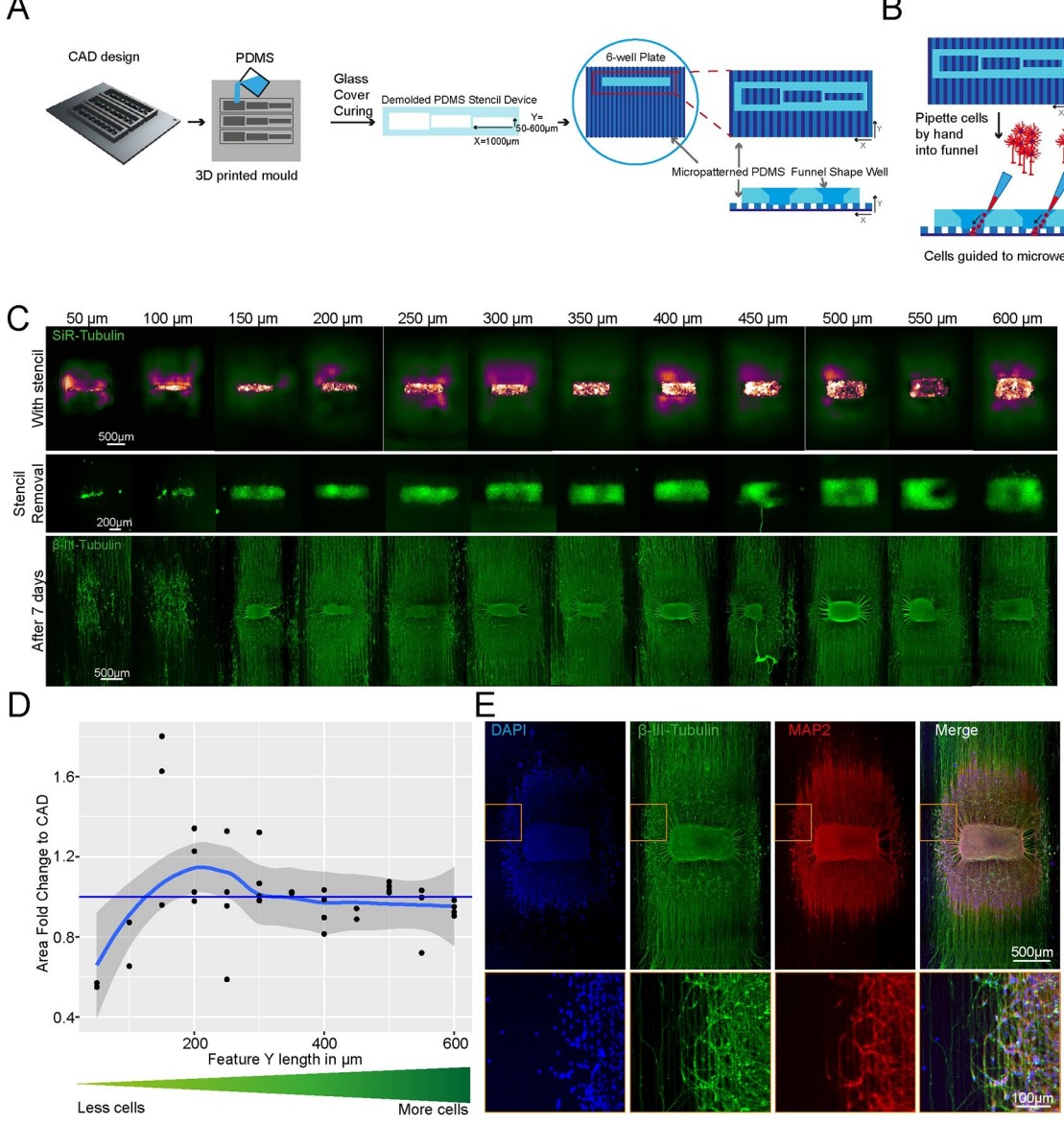

**Fig 2. SLA 3D printing enables control over cell location and number in an open well.** (A) Schematic overview of the investigation dry plating strategy to combine PDMS casts from 3D-printed moulds with PDMS microgroove substrate to control cell body location and number. Well sizes range from 600 μm × 1,000 μm to 50 μm × 1,000 μm in 50 μm intervals. (B) Schematic overview of funnel shape well for easy manual seeding in microwells. (C) Representative images of stencil devices filled with pre stained (Silicon Rhodamine-tubulin) MN progenitors with device still in place (top), after device removal (middle), and axonal β-III Tubulin following fixation after 7 days of culture (bottom). (D) Comparison of seeded cell area after stencil removal to CAD specified values by fold change. (E) Representative images of 3D aggregoid with 2D axon elongation stained with DAPI (first image), axonal β-III Tubulin (second image), and Dendritic MAP-2 (third image) simultaneously (fourth image).

into micro-sized wells in a suitable volume of medium to avoid excessive evaporation and cell death (Fig 2A and 2B).

PDMS stencils from these 3D moulds allow seeding by "dry plating," whereby a stencil is placed in a dry conventional tissue culture plastic vessel and cells in suspension are manually pipetted in the stencil device, isolating the cell bodies from the residual well and allowing them to adhere at these specific positions. With the cell bodies secured, the stencil device can be

removed and the whole well filled with culture medium, while the adhered cells remain in their specified position. For this "dry plating" process, a strong fluidic seal surrounding the PDMS stencil wells is necessary, requiring a flat surface between the stencil and substrate below. Without specific steps to adjust the surface roughness of prints, PDMS casts from 3D-printed moulds are inherently rougher than those from micropatterned silicon wafers used for casting microfabricated PDMS devices (**S7 and S9** Figs). We, therefore, implemented an additional clamping step before PDMS curing, using a silanised glass slide (see M&M) to cover the PDMS surface, which is in contact with air, taking advantage of the flat surface provided by the glass (**S10A Fig**). We evaluated the efficiency of clamp-cured stencil fluid seals when placed on a PDMS micropatterned surface with $10 \times 10$ µm-grooves using a blue dye. An effective seal was achieved in all stencils cured using the additional clamping, denoted by dye reaching the microgroove substrate in the well area only and spreading within these specific grooves. Stencils cast without clamping showed uncontrolled dye spreading throughout the devices, verifying a lack of fluid seal (**S10B Fig**).

We were then able to use the optimised stencil devices to answer a biological question and investigate the minimum number of iPSC-MNs required to form a self-organised 3D neural aggregate on microgrooves for axonal elongation, a process determined by chemotaxis and topography. To achieve this, we used the above-described stencils with a funnel-shaped reservoir and rectangular wells, varying in Y-dimension to reduce the stencil well size and control cell amount. The well dimensions were homogenous and faithful to CAD specifications throughout the print sizes down to 50 µm in Y (**S11 Fig**). These PDMS stencil-well devices were placed on the extra cellular matrix (ECM) coated and dried micropatterned surface with axonal guidance grooves [42], and the iPSC-derived MN cell suspension was manually pipetted into the dry wells of the device. To avoid potential air pockets in the smaller wells, as it is common for non-functionalised PDMS, we performed oxygen plasma treatment on stencils prior to cell "dry plating" (**S12 Fig**). Compact rectangular "aggregoids" (i.e., 3D cell clusters generated by reaggregating single cells from a culture) with decreasing size were achieved during seeding and were maintained following device removal. Staining with β-III-tubulin after 7 days in differentiation medium revealed that wells with a size of down to 150 µm provide suitable cell numbers for aggregate formation. However, the 2 smallest well sizes did not provide the environment for aggregate formation and cells migrated across the topography (Fig 2C and 2D). Subsequent staining with compartment-specific markers showed a clear separation between dendrites and axons in the open well devices of compact aggregoids (Fig 2E).

In summary, stencil-well devices cast with SOL3D can be used to control cell location in an open well, facilitate control over different cell numbers in the same device, and enable cell compartment-specific investigations.

## Spatial, temporal, and morphological control over cell–cell interaction using tailored plating devices

We next explored the potential to use SOL3D for more complex cultures, incorporating different cell types and plating time points.

First, to plate multiple cell populations within the same devices, we sought to take advantage of PDMS natural hydrophobicity coupled with the large rectangular well design and funnel-shaped well profile showed in Fig 2A. The high contact angle between media and hydrophobic PDMS allows to achieve confinement of the different cell suspension droplets, which generates enabling complete fluidic separation between adjacent wells containing different cell populations, even with manual plating. We sought to utilise this strategy to simultaneously plate different iPSC-derived MNs populations within the same tissue culture well in different spatially

separate pockets of the PDMS device. For this, we used fluorescent RFP$^+$ and untransfected MNs, which we plated manually within adjacent pockets in the same device placed on an ECM-coated well of a 6-well plate. The different cell populations were left to adhere for 2 h before the plating device was removed. After removing the device and further cell culture for 72 h, all MNs were stained with a silicon rhodamine tubulin dye (here depicted in green for visualisation), to visualise all neurites and cell bodies. A line graph analysis across the whole device showed that all wells contained MNs (RFP$^+$ and RFP$^-$/Tubulin$^+$) and in every second well, RFP$^+$ cells were present (Fig 3A and 3B), demonstrating multi cell type seeding in confined predetermined spatial groups within a single device.

We then sought to further increase the complexity of our in vitro cultures by seeding multiple cell types at different time points within the same well, taking advantage of the efficient and reversible fluid seal between our devices and the culture plate. For this, we placed 2 rectangular plating devices, approximately 2 mm apart from each other on an ECM-coated micropatterned surface within a well of a 6-well plate, as described above. Initially, one device was used to dry plate iPSC-derived cortical neurons [43] and astrocytes [44] in a 1:1 ratio and removed after 24 h, while the other device was kept empty. The whole tissue culture well was then filled with differentiation medium and cultured for 9 days. During this time, cortical axons guided by the microtopography extended toward the empty device, which maintained its initial fluidic seal even surrounded by medium. On day 9, GFP$^+$ MNs were seeded in the second device by first lowering the level of the medium within the well to be below the edge of the plating device, and then seeding the MNs in suspension directly within it. After allowing for cell attachment, the second device was also removed, the well refilled with fresh medium and cells cultured for further 9 days. The position of the different cell types was then verified using immunocytochemistry (ICC) for astrocytes (GFAP) and neurons (both MNs and cortical neurons, β-III-Tubulin), as cortical neurons could be identified by the overlap of GFP and β-III-Tubulin. Using these tailored removable SOL3D-generated plating devices, we were able to easily plate 3 different cell types at 2 different time points within the same culture well, creating a complex neural circuit and demonstrating true spatiotemporal control over cell seeding in a cost-effective and highly adaptable fashion (Figs 3C, 3D, and S13). Additionally, we demonstrated also multiple time point seedings within the same well using large format "nesting" plating devices that can be used to construct large-scale cell and tissue arrangements (S14 Fig).

Next, we tested whether the ease of available prototyping using our optimised SOL3D protocol could enable investigation and manipulation of the fundamental behaviour of complex iPSC-derived MN cultures. It has been shown that aggregation of cells using different geometries has an influence on the signalling environment and patterning of aggregates [45]. We therefore designed and fabricated moulds for PDMS stencils with 3 different well shapes: rectangular, circular, and triangular, to create geometrically constrained neural aggregates. With the advantage of producing multiscale features simultaneously, we were able to preserve the funnel reservoir and straight well design from previous moulds (Fig 3E). Using these multi-shaped stencils, we seeded motor neuron progenitors (MNPs) as before on our micropatterned substrate and allowed axons to extend for 11 days (Fig 3E). Here, MN aggregates maintained faithful area and aspect ratios to CAD specifications on day 2 after stencil removal. After 11 days, the groups also retained their specific geometry although showed slight changes in the aspect ratio and area over time (Fig 3F–3H). Next, we used these PDMS stencils on nonpatterned and uncoated tissue culture plastics to avoid cell adherence, directing self-organisation of aggregate-like structures (S15A Fig). Here, we seeded cortical progenitors in Matrigel and were able to generate differently shaped aggregates demonstrated by SiR-tubulin live dye images after 24 h (S15B Fig). SOL3D fabrication can therefore be used as a valid method of fabricating constructs for controlling cellular interactions in complex cultures of multiple

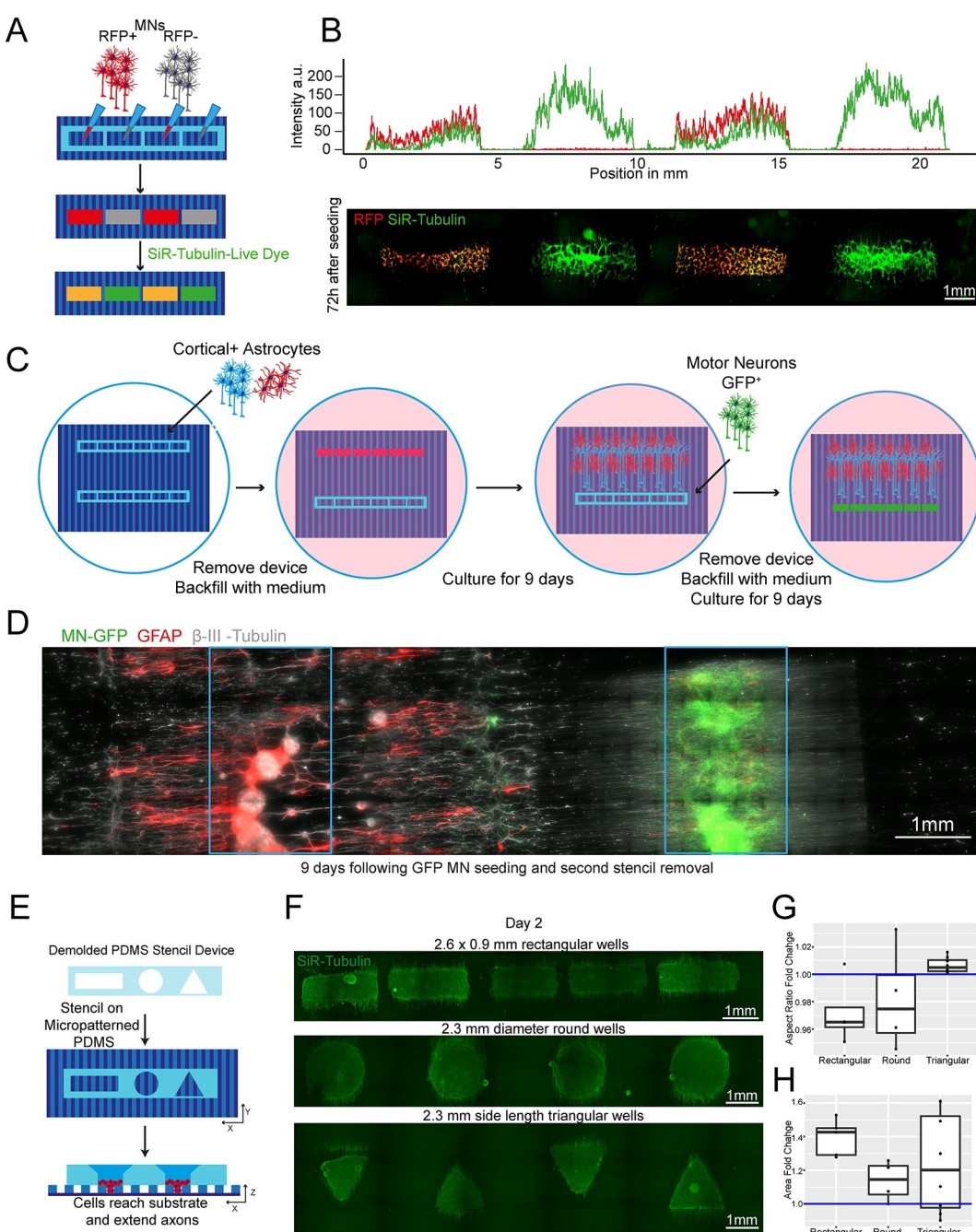

**Fig 3. Plating devices enable spatiotemporal control of cell plating with different geometries for construction of complex neural circuits.** (A) Schematic overview of alternate seeding of RFP and non-RFP+ motor neurons in the same device and the following live cell staining. (B) Representative line profile of stained across the well showing segregation of individual populations to their designated wells—RFP+ only in wells 1 and 3, but SiR-tubulin+ (here in green) in all wells (top). Representative fluorescence images of stained RFP+/− motor neurons (bottom). (C) Schematic overview of the multi-device protocol for constructing a neural circuit using 2 stencil devices and 3 different cell types (MNs, cortical, and astrocytes) with seeding performed at different time points. (D) Composite of the complete circuit after 19 days of culture. GFP transfected motor neurons = green, Glial Fibrillary Acidic Protein (GFAP) identifies astrocytes, β-III–Tubulin identifies cortical neurons and the tubulin in GFP+ motor neurons. Blue device well shapes overlaid for illustrative purposes. (E) Schematic overview of protocol for manipulating aggregate geometry in combination with existing microgroove. (F) Representative images of SiR-tubulin live-cell stained motor neuron aggregoids at day 2. (G) Boxplot of aggregoid aspect ratio fold change by shape between CAD (blue line) and day 2 of culture from β-III-Tubulin channel (top). (H) Boxplot of aggregoid area fold change by shape between CAD (blue line) and day 2 of culture from β-III Tubulin channel. Data points for G and H can be found in the files 3G-Data and 3H-Data in S1 Data.

geometries for both 2.5 (i.e., partially tridimensional adherent cultures) and 3D non-adherent cultures (e.g., aggregates) depending on the seeding substrate.

## 2.2 SOL3D fabrication allows the generation of micro topographies

In the initial experiments presented in Figs 2 and 3, we employed SOL3D-fabricated devices on top of microgrooves generated with conventional photolithography. As this method is not available to all labs, we then focused on achieving the same level of organisation within the culture but on a platform purely based on SOL3D-manufactured devices. While the 3D vat polymerisation printers do not have the same resolution as photolithographic equipment, we could not simply recreate the 10 μm grooves.

Moreover, with current LCD-based illumination, repetitive patterns in close proximity to each other and close to the minimal resolution pose a challenge for 3D vat polymerisation printing, due to the illumination pattern and diffraction of the light **(see supplementary guide page 15 in S1 Text).**

However, the key parameter is the biological organisation rather than the material geometry, and we, therefore, focused on obtaining a design that can both be printed with SOL3D and achieves the same neurite orientation. We developed a design with different groove geometry and dimensions (Fig 4A), with a groove depth of 200 μm and width of 100 μm. Optical profiling of the 3D prints reveals that the grooves are shallower than the CAD design, around 60 μm, and less wide, as expected (**S16 Fig**). Despite these limitations, we tested if the topography is sufficient to align axonal elongation. We plated MN neural aggregates [42] in a SOL3D-manufactured stencil device on SOL3D-manufactured grooves following the "dry plating" protocol (Fig 4B). Visualisation of axons (β- III-Tubulin) after 7 days of culture revealed an alignment with the topography throughout the axonal length (Fig 4C and 4D).

Another process benefitting from alignment to topographies is the formation of myotubes [46]. We seeded myoblasts on the SOL3D grooves and a flat control surface (Fig 4E). Cells were differentiated for 3 days and then visualised using SiR-Tubulin. Myoblasts cultured on the patterned surface show alignment to the topography, compared to the control conditions where cells are randomly positioned (Fig 4F). While the cells follow the given topography, the differences in size and shape of the SOL3D grooves compared to the photolithography pattern, might evoke different interaction and biological responses, making the 2 different groups not entirely comparable, but Sol3D manufactured grooves provide a suitable alternative to microfabricated grooves and provide guidance and cell alignment in some circumstances. As a resource, the SOL3D grooves provide a suitable alternative to microfabricated grooves and provide guidance and cell alignment. Taken together with the SOL3D stencil-like devices, a whole on-chip platform can be generated using SOL3D.

## 2.3 Customisable SOL3D fabrication as a tailored alternative to standardised commercially available culture platform

The ability to create customisable devices and substrates suitable for cell culture or other biological experiments, with μm to cm sized features, in a fast, reliable, and cost-efficient manner would be particularly useful in any wet lab, granting independence from high costs, delivery times, and availability of the equivalent commercial products, while enabling substantial customisation. For example, most cell culture vessel layouts are standardised and not tailored to the need of an individual laboratory or a specific cell type, causing higher costs and potential compromises in experimental setups. We therefore aimed to test whether our optimised SOL3D mould protocol could be used to reproduce and further customise relevant features from popular commercially available cell culture products. These

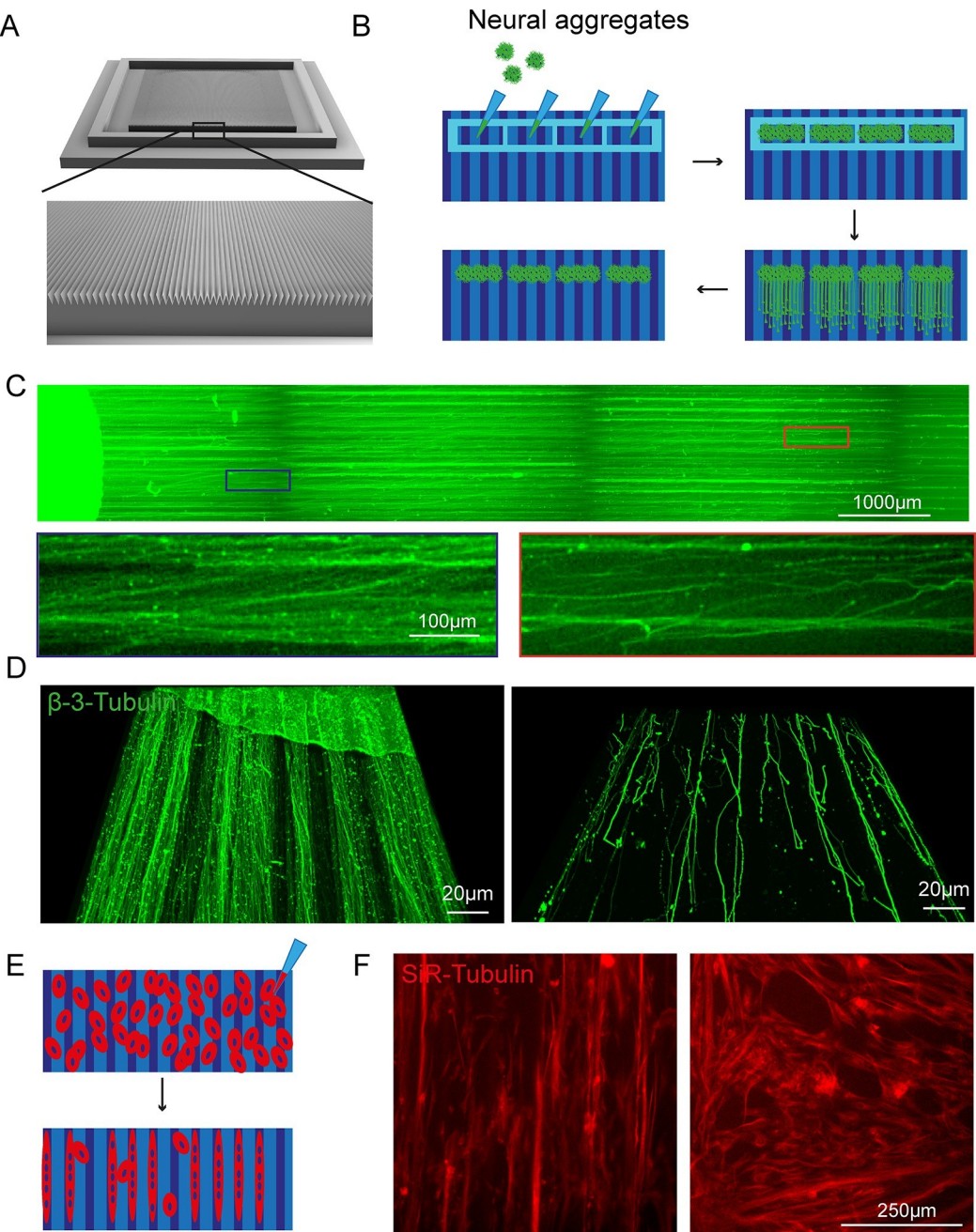

**Fig 4. SOLID fabricated grooves guide axonal elongation and alignment of muscle fibres.** (A) CAD design of triangular grooves 100 μm wide and 200 μm deep with 75° triangular spacing in between. (B) Schematic overview of "dry" neural aggregate seeding in PDMS stencil devices on the 3D-printed grooves and axonal elongation. (C) Representative overview image of axons (β-III-Tubulin) on the SOLID fabricated grooves with magnifications of axons aligning with the topography in the proximal (blue) and distal (orange) compartment. (D) Representative 3D reconstruction of axons (β-III-Tubulin) on grooves in the proximal (left) and distal (right) compartment. Across compartment axonal alignment to the topography is given. (E) Schematic representation of myoblast seeding and differentiation on SOLID grooves. (F) At day 3 of differentiation myotubes (SiR-Tubulin) align to the given topography (left) compared to cells cultured on a nonpatterned surface (right).

constructs can be customised in dimensions and/or shapes for individual experimental aims, while remaining cost-effective, highlighting the versatility and accessibility of our system to enhance biological investigations.

## PDMS bonding for chamber slide devices

Chamber slide systems and other microscopy-ready hybrid culture devices are commercially available systems that allow cells to be cultured within neighbouring wells directly on cover slides for high-resolution imaging, providing small well sizes and imaging-compatible set-ups for high throughput and convenience. To ascertain if our SOL3D protocol could be applied to mimic these constructs, we designed a chamber slide system that can be permanently bonded to an imaging coverslip either using oxygen plasma treatment or a UV-sensitive resin adhesive. Importantly, the adaptation of our construct for use with UV resin makes this method accessible to labs without a plasma cleaning system (Fig 5A). Our design was fabricated using the SOL3D protocol and was size matched to a 60 mm × 24 mm microscopy coverslip with 12 circular wells with funnel shapes. As described above, we generated a fluidic seal by clamping the device with a glass slide during PDMS curing to isolate neighbouring wells. This extremely flat PDMS surface allows fusion of PDMS to the glass slide using oxygen plasma bonding, or the simple application of a UV adhesive. It is important to note that the UV adhesive is resin based and therefore cytotoxic and cannot be used on any medium-facing area. Astrocyte progenitors were then seeded into selected wells at different concentrations (Fig 5B). Staining with a live dye (SiR-Tubulin) revealed an intact fluidic seal in both devices, liquids maintained in the respective wells, and healthy astrocyte progenitor populations. We further demonstrated the high-resolution imaging compatibility (Fig 5C), as cells are seeded on a glass slide. Ostensibly, we have demonstrated that both oxygen plasma and UV resin are suitable for PDMS bonding of a chamber slide device and highlighted the capabilities of 3D printing for the fabrication of bespoke chamber slides in a fast and cost-effective way. With the recent release of the Phrozen Mini 8k, with increased resolution, we sought to test our protocol for also this printer and the provided resin. Microfluidic devices are expensive and single use only, with no customisation opportunity, while they require a flat surface and a high grade of detail. We used a triple chamber design (S17A Fig) with channels of 100 μm in width and only 10 μm in height, a z-layer thickness that is barley achievable with the 4k Mini, even with optimised parameters. To further prove the advancements in printing quality, we did not clamp a glass slide onto the print for a flat surface but directly used the printed surface. Indeed, the channels were 9.4 μm in height and around 110 μm in width (S17B and S17C Fig). After oxygen plasma treatment, the surface was sealed onto a glass slide and a fluorescein solution was flowed through using fitted tubing and a syringe. Time-lapse images clearly show that the channels are connected and the solution is washed in, without leakage (S17D–S17F Fig). Overall, we confirmed that our pipeline works with the next-generation printer and that with improved quality of the prints fine grade details can be achieved also in the z axis, as well as a smooth print that does not require glass clamping.

## Custom microwell arrays for embryoid body formation

The first design we tested for this purpose was an array of pyramidal-shaped microwells (390 × 350 × 150 μm) that we fabricated using the optimised protocol with no coating step, as it is required for this small feature size (<500 μm) (Fig 6A and 6B). These microwells have become essential for induction of specific cell lineages from iPSCs and for aggregate research [47,48]. One of the most important functions of these wells is to ensure homogenous aggregate size for reproducible results, for example, generating embryoid bodies (EBs) of regular size

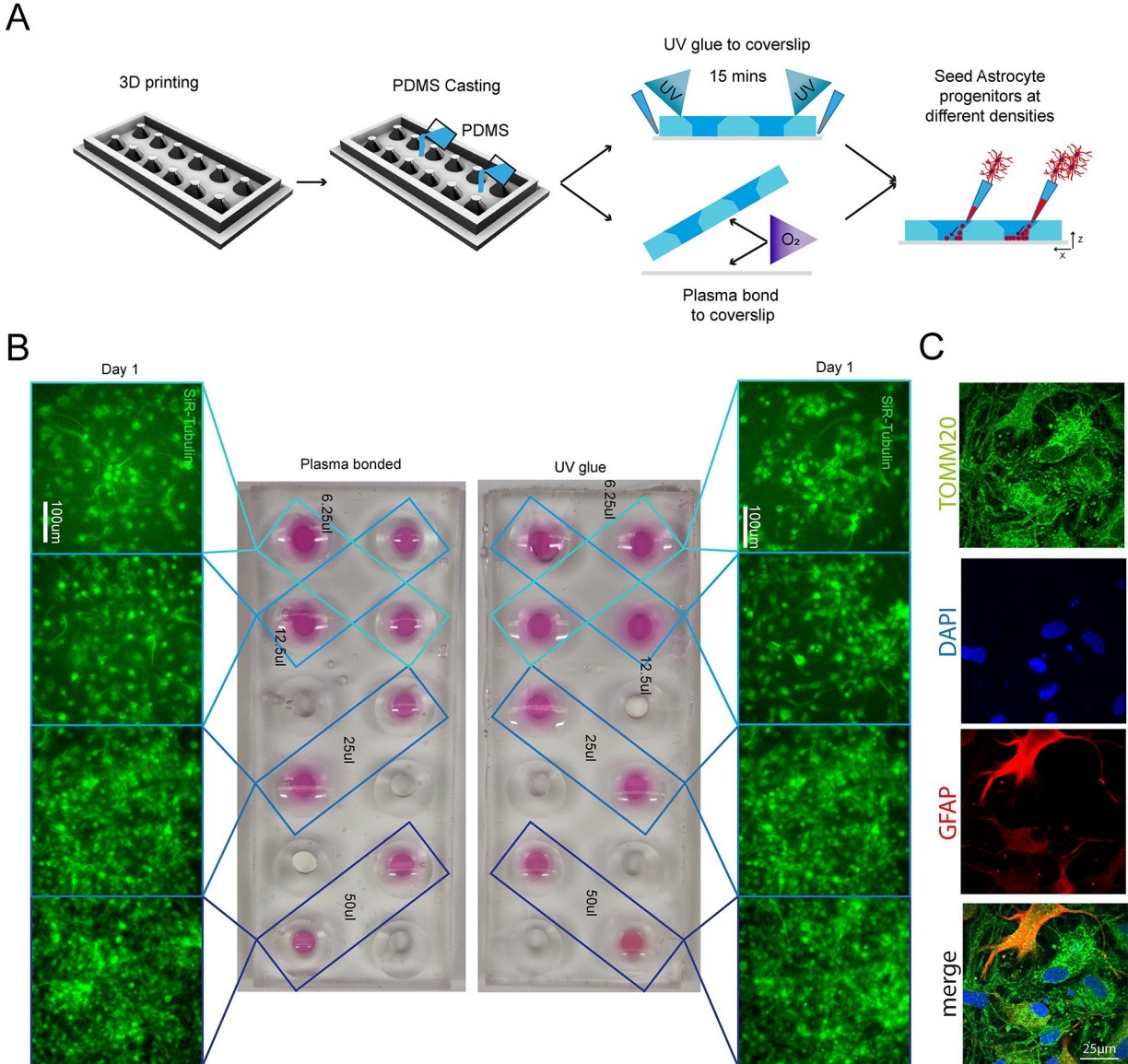

**Fig 5. 3D printing can create fully customisable imaging chambers with complex well geometry suitable for cell culture via 2 different methods of PDMS bonding.** (A) Schematic overview of design and manufacture of chamber slide device with large wells with (1) UV glue or (2) oxygen plasma bonding to a glass coverslip to seal the wells. (B) As demonstration of the seal quality and viability for cell culture astrocyte progenitors were seeded in different densities in nonadjacent wells and cultured in chamber slide device. Representative images of SiR-tubulin live dye-stained astrocyte progenitors 1 day after seeding in chamber slides bonded with different methods. (C) High-resolution imaging of astrocytes (GFAP) and mitochondria (TOMM20) cultured in custom-made SOL3D PDMS chambers.

and shape. We used our moulded microwell arrays to form EBs from an iPSCs suspension (Fig 6C) and quantified their size after 4 days of culture on the devices. In our microwells, iPSCs formed EBs with consistent diameters, verifying the suitability of our custom PDMS moulds to create small regular arrays of features (Fig 6D). The microwells generated by our protocol are therefore suitable for generation of homogenous EBs with the benefit of substantial customisation of well shape and size at a low cost.

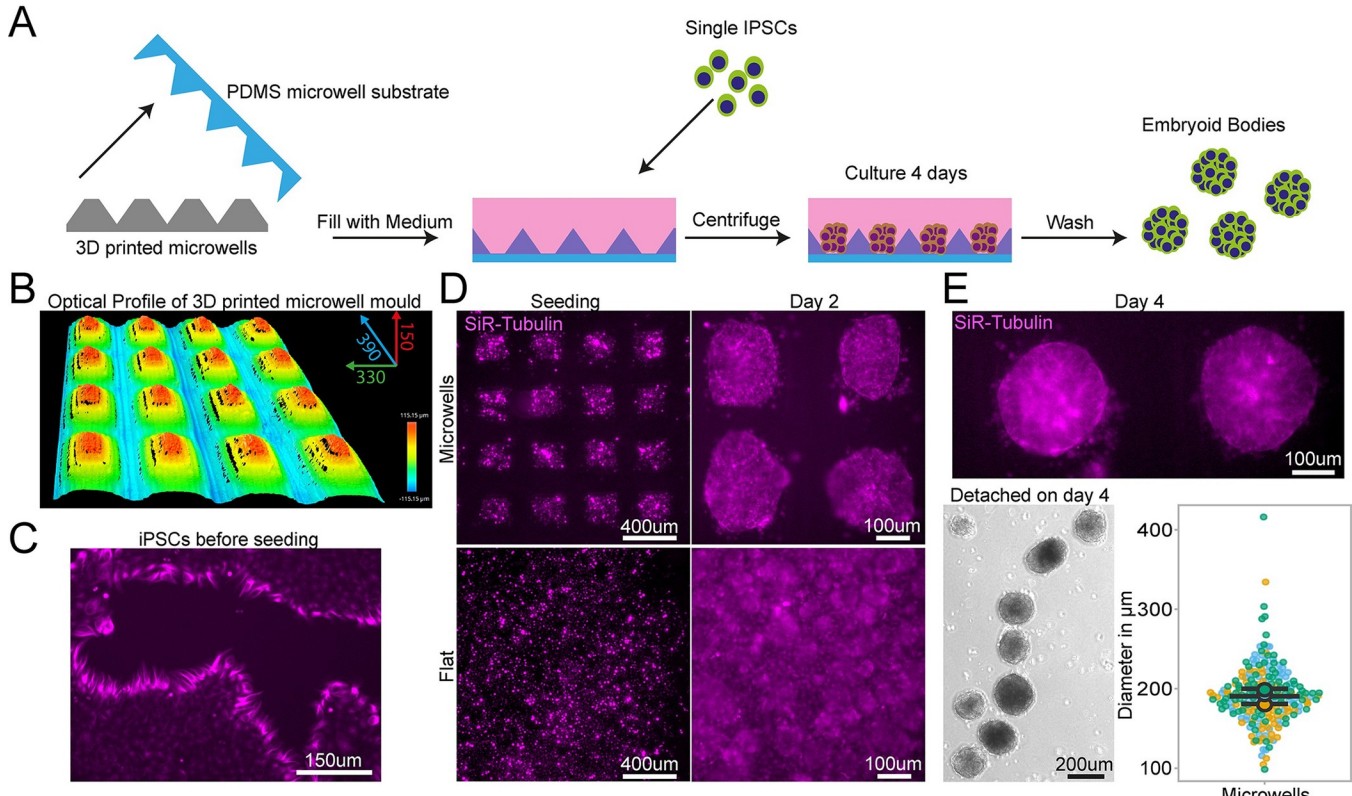

**Fig 6. PDMS substrates cast from 3D-printed devices permit regular-sized embryoid body.** (A) Schematic of design, manufacturing and seeding of IPSCs on microwells. (B) Representative optical profile of 3D-printed microwell device with well sizes of 390 μm length × 350 μm width × 150 μm height. (C) Representative SiR-tubulin live cell dye images of IPSCs before seeding in microwell mould cast. (D) Representative SiR-tubulin live cell dye images of IPSCs seeded on PDMS cast from 3D-printed microwell device compared to flat PDMS substrate after seeding (left) and 2 days culture (right). (E) Representative SiR-tubulin image of fused embryoid bodies on microwell PDMS mould prior to detachment after 4 days in culture (top) and embryoid bodies following washing off the PDMS microwell substrate (bottom left). Quantification of embryoid body diameter detached from microwell PDMS mould demonstrates homogenous size of embryoid bodies (bottom right). Data points can be found in the file 6E-Data in **S1 Data**.

## Large tissue engineered devices with complex designs

Generating complex devices for tissue engineering often combines relatively small features within large constructs and has so far proven challenging to implement in most laboratories, as construction processes are complex and time consuming, requiring dedicated expertise. Most devices of this kind are therefore sourced from commercially available suppliers, with limited possibility of customisation and at a high cost. For example, tissue-engineered 3D muscle constructs use a variety of devices for suspending large cell laden hydrogels during culture using thin suspension posts [49,50]. They are comprised of small pillars with complex shaped end feet, which serve to suspend the hydrogel construct and provide mechanical stiffness to aid differentiation. As these posts are difficult to manufacture and arrive pre-made of a single size and shape, no customisation is available, e.g., miniaturisation or altered substrate stiffness. Successful 3D adaptations have been implemented for smaller and less complex muscle post, however, with low ease of production for 3D SLA printing [51].

We used our SOL3D protocol to fabricate a device for suspended 3D muscle culture with customisable post size and overall dimensions. The challenge, in this case, stems from the fact that these devices do not have large flat faces, they present thin complex features and need ideally to be produced as a single component to avoid complex assembly steps that can introduce

variability. A single mould system would in this case not be sufficient, as the lack of air in contact with the complex shapes would prohibit the successful demoulding of the structure. We created a two-part mould/injection system using SOL3D, which can easily be assembled by clamping for curing after PDMS is poured into the mould. Optimisation of the moulds showed that an unequal distribution of the design between the 2 parts (70/30) is beneficial for successful demoulding, resulting in a reproducible single device with the desired dimensions, in this case twice as large as the commercial alternative—a 2 cm muscle compatible with 12-well plates (Fig 7A). We compared our 12-well plate 3D posts to the commercially available 24-well adapted equivalent (see M&M, Muscle culture), using immortalised myoblasts.

Following our protocol to generate 3D bioengineered muscle [50,52], we first created a pouring mould by filling liquid agarose around a 3D-printed rectangular spacer that was removed after the agarose has set (**S18 Fig**). Subsequently, myoblasts were seeded in fibrin hydrogels within the agarose mould, and the SOL3D-fabricated posts (or commercially available devices [53] used in Maffioletti and colleagues) were inserted within the still-settling fibrin constructs [50]. After 2 weeks of differentiation, we performed electrical micro stimulation to measure muscle contractility—a hallmark of successful 3D muscle culture—on both constructs at 20 mV with 0.5 Hz frequency (Fig 7B), which showed periodic contractions for both SOL3D and control devices. Immunostaining of the muscle tissue showed the presence of terminally differentiated myosin heavy chain (MyHC) and titin positive multi-nucleated fibres in both constructs. Directionality analysis revealed that myofibres were preferentially aligned along the posts (Fig 7C and 7D). After demonstrating the suitability of our 3D posts for muscle cell culture, we focussed on miniaturisation of the posts, to enable obtainment of the same biological outcomes with fewer materials. For this purpose, we designed and manufactured insets for 24- and 48-well plates with the SOL3D protocol (Fig 7E). However, in addition to changing the dimensions, we also utilised the benefits of the tuneable stiffness of PDMS by changing the ratio between monomer and curing agent, while remaining optically clear and retaining most of its surface properties. Here, we used the 24-well plate format moulds and polymerised PDMS in its standard formulation (1:10) and in a lower ratio of the curing agent (1:20). The lower the ratio of curing agent to PDMS the softer the material will be, and to confirm that we can also use softer PDMS in our 3D Sol3D moulds, we measured the modulus of the muscle devices using compression (Fig 7F). As expected, the storage and loss modulus are higher for 1:10 PDMS compared to the 1:20 formulation (Fig 7G).

In summary, our protocol allows for complex and scalable features to be easily moulded in PDMS with the additional benefit of customisation in all aspects of design for improved function of 3D engineered muscle tissues.

## 2.4 Entirely SOL3D-fabricated hydrogel moulding, culture, and imaging system

Hydrogel cultures offer a great opportunity for in vitro 3D modelling, as they are easy to produce, can be tailored to specific applications and provide an in vivo-like environment with a complex architecture, and mechanical properties that more align with specific tissues. The low stiffness is a particular advantage for neuronal cultures as the brain is one of the softest matters in the human body, but it also creates challenges in the handling of these hydrogel cultures. Bioprinters can be used to directly form hydrogel cultures with specific shapes and positions of cells within the gel; however, due to the complexity and engineering effort required, accessibility is limited for many labs. Moreover, the use of 3D cultures also presents challenges in adapting conventional culture vessels and plates to the specifics of the hydrogel constructs.

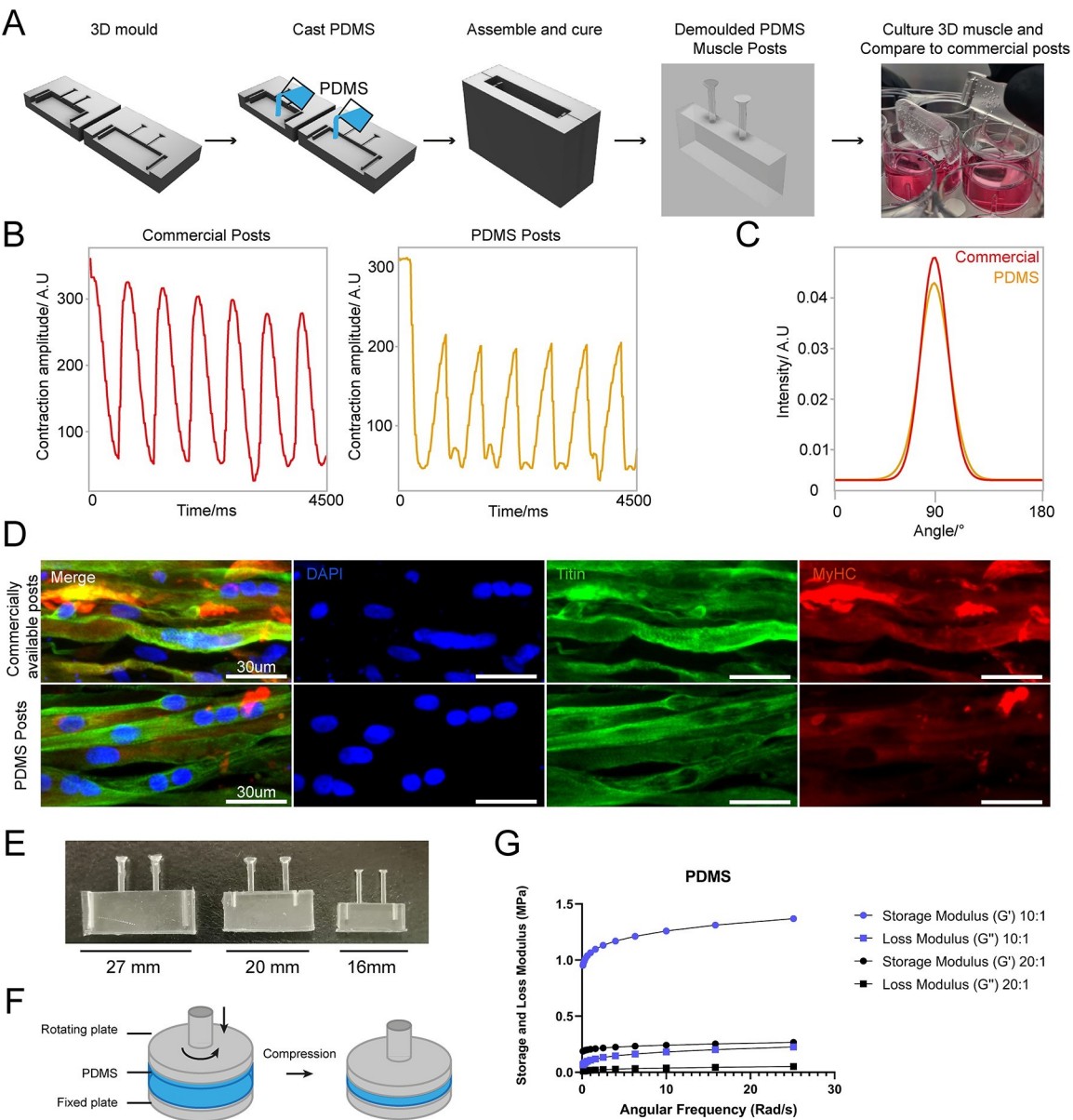

**Fig 7. 3D sandwich moulds for PDMS casting to generate complex cell culture devices.** (A) Schematic overview of design and PDMS casting strategy for a 3D sandwich mould. Immortalised myoblast hydrogels were then formed around PDMS posts, differentiated and cultured for 2 weeks. (B) Comparative contractility analysis with microstimulation at 20 mV with 0.5 Hz frequency of differentiated 3D muscle between crafted PDMS posts and commercially available posts. (C) Directionality analysis of fibre alignment in differentiated 3D muscle fibres between PDMS posts and commercially available posts after 2 weeks differentiation. (D) Representative images of myoblast differentiation (Titin) and developmental stage (MyHC) on PDMS posts and commercially available posts after 2 weeks differentiation. (E) Images of posts with different dimensions suited for 12-, 24-, or 48-well plate (from left to right). (F) Illustration of compressing PDMS to measure the modulus through a lateral and rotating movement. (G) The Storage and Loss Modulus of PDMS with different formulations (1:10 or 1:20) at changing angular frequency.

We sought to provide a 3D culture system for hydrogels that allows ease of handling and culture of hydrogels, is entirely customisable and cost-effective, requiring no more than the equipment needed for SOL3D to be implemented.

We started by designing and manufacturing a PDMS mould for shaping hydrogels. This mould is manufactured using the SOL3D protocol and our clamping system for a flat bottom

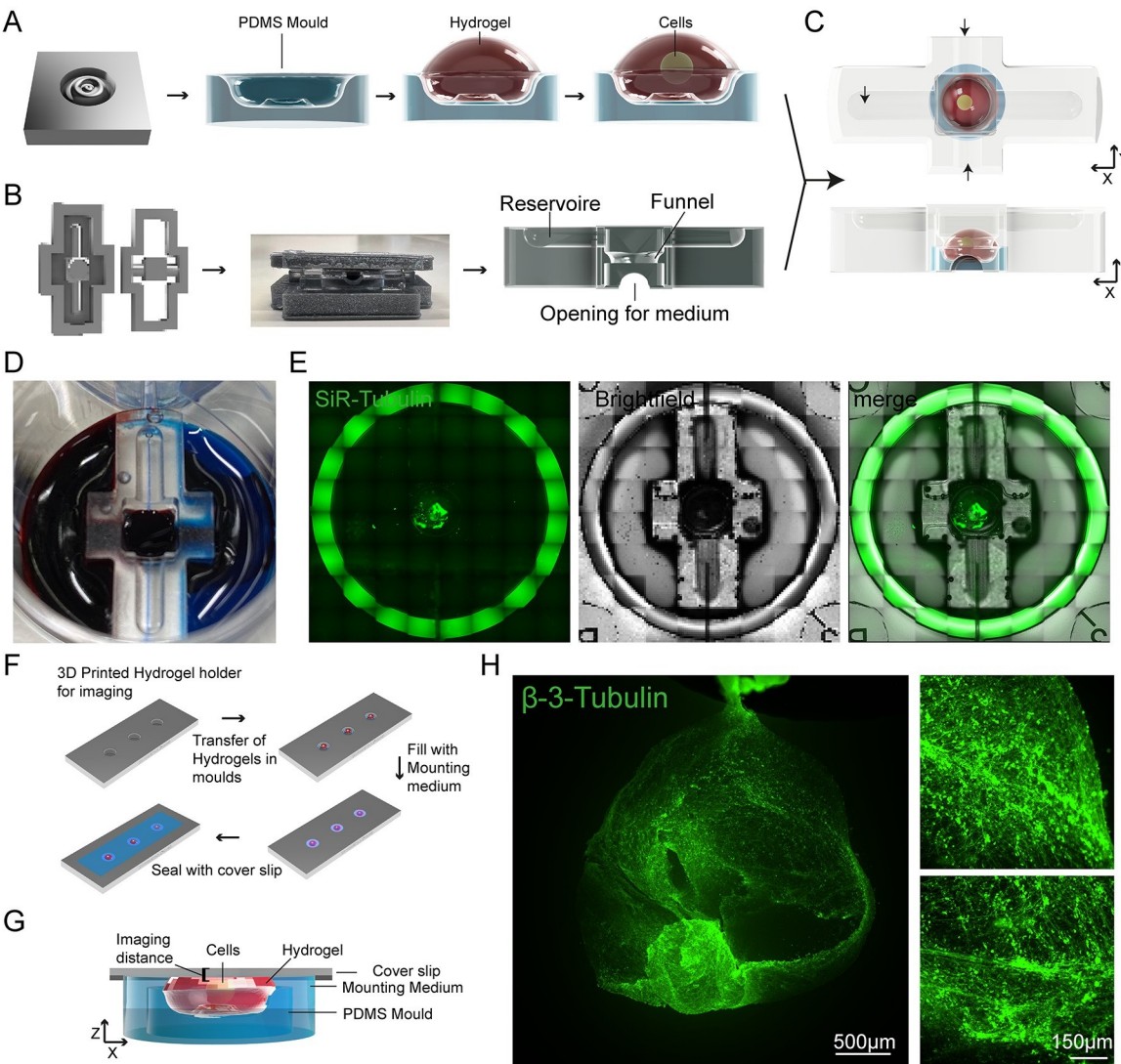

**Fig 8. Entirely SOL3D-manufactured hydrogel moulding and culturing chamber system with customised imaging chamber.** (A) Schematic overview of PDMS mould generation for hydrogel moulding and MN seeding. Cells are manually pipetted into a preformed hydrogel moulded in the SOL3D-fabricated PDMS mould. (B) Schematic overview of the PDMS chamber manufacturing process using SOL3D. The chamber has a complex design with a funnel shape within the structure and multiple openings. We used a two-part design for ease of demoulding. (C) Schematic overview of the combined hydrogel with MNs placed in the mould (A) and a diffusion chamber around the construct (B), arrows indicate the flow of medium. (D) Optional media compartments highlighted using food colouring (blue, red) and PBS (top). (E) Representative images of the SOL3D chamber system (Brightfield) with MNs (SiR-Tubulin) in culture in a 6-well plate. (F) Schematic overview of the mounting process of the complex hydrogel samples for fluorescent imaging. A SOL3D manufactured holder allows the transfer of whole constructs without disturbance and mounting close to the coverslip for imaging. (G) Schematic overview of the hydrogel within the holder. The PDMS mould sits in the bottom of the well with the hydrogel and cells on top. The sample is surrounded by mounting medium and covered with a thin glass coverslip for imaging. Due to the custom holder system, the distance between sample and glass is minimised allowing for fluorescent imaging. (H) Representative fluorescent images of MNs cultured for 7 days in the SOL3D hydrogel chamber system and mounted in the SOL3D chamber. MNs (β-III-Tubulin) seeded in the hydrogel extend many axonal processes.

surface so that the mould stays at the desired position. Once placed, pipetting the hydrogel into the mould and seeding of MNs progenitors is performed manually ([Fig 8A]). To further avoid detachment of the gel from the mould and control medium flow to the construct, we developed a chamber system that can be placed on top of the hydrogel with the mould. The

design includes a chamber for the hydrogel and mould that is connected with a funnel to a reservoir on the top. This restricts the movements of the hydrogel and at the same time offers the opportunity to provide medium from the top (Fig 8B). The chamber has 2 round-shaped openings that provide a flow of medium from the respective side of the well (Fig 8C). We also matched the size of the chamber to the size of a 6-well plate, this allows for different media to be added from both sides and the top of the device (Fig 8D). We used IPSC-derived MNs in a custom-made hydrogel and cultured the construct for 7 days (Fig 8E). Visualisation of axonal outgrowth in the hydrogel requires fluorescent imaging with a higher magnification. The size of hydrogel constructs limits the simplicity of imaging approaches and usually requires more complicated and expensive setups. We designed an imaging holder which is custom fitted to the size of the hydrogel mould. After staining for axons (β-III-Tubulin), the hydrogel chamber is removed and the mould containing the gel can be easily transferred to the holder. Due to the tailored dimensions, only a small amount of mounting medium is required to cover the sample and then a cover slip is placed on top of the holder (Fig 8F). This results in a very tight mounting of the sample, where MNs are close to the glass slide, and minimal disturbance of the sample itself, maintaining the complex architecture (Fig 8G). Fluorescent imaging with a low magnification (4×) shows the overall structure of the MN aggregate, and higher resolution images (20×) show that MNs extend long axons in the hydrogel construct (Fig 8H). This SOL3D-fabricated hydrogel moulding, culturing, and imaging system provides an easy and inexpensive system that is not limited to hydrogels but can be adapted for any 3D system.

## 3. Discussion

In this study, we have developed a fast, high-resolution, and cost-effective protocol to quickly prototype highly versatile cell culture compatible devices that can be applied to a range of different applications, from live cell imaging to microfluidics and even advanced tissue engineering, which we have called **soft-lithography with 3D resin moulds using vat polymerisation (SOL3D).** This methodology allows any lab, even those with very minimal prior expertise in the field or without dedicated resources, to effectively set up a complete microfabrication prototyping system and produce culture devices tailored to their specific biological experiments with minimum expenditure.

The widespread commercialisation of 3D printing has led to a significant development of resolution and accessibility, which accompanied by free repositories and software packages, have significantly lowered the entry barrier for the adoption of this technique. As a result of this rapid development and the sheer number of new resins and printers becoming available, it is difficult to get an overview of the suitable protocols and materials for a given application. A general protocol for printing and fabrication using any commercially available product is therefore highly desirable.

We have capitalised on these technological and community developments to overcome one of the primary barriers to complex microfabrication in a biology-focused lab by developing and testing a robust pipeline of fabrication for 3D vat polymerisation, post processing and PDMS casting, which enables for complete customisation of any cell culture device without further establishment or optimisation. We tested resins from various manufacturers and identified one suitable for PDMS casting and high-resolution prints. In particular, one of the resin compositions performed optimally without the application of a paint layer for high-resolution prints. This feature allows users to exploit the full potential of high-resolution prints, making ink coating for small details redundant [35]. Interestingly, this resin was originally developed for high-resolution printing and has a noticeably lower viscosity than other resins we tested, parameters to consider when evaluating different resin compositions for this method.

However, enclosed prints such as the large tissue culture constructs (Fig 7) would still not be able to cure sufficiently without a coating layer that isolates the PDMS from the mould. We also tested Resin A with a different commercially available mid-price SL printer and found no observable difference in print quality, PDMS curing, or biocompatibility, showing easy transfer of optimised parameters for resins to multiple printer systems and designs.

When considering microfabricated devices for biological experiments, especially with cell culture and other in vitro set-ups, the topographical features one might want to add can range roughly from subcellular scale (<5 μm, e.g., nanoindentations [54] and other nanostructures [55]), cellular scale (10 to 100 μm, e.g., microgrooves [56] and other microwells [57]), multicellular (200 to 1,000 μm, e.g., microfluidic channels [58]), or tissue scale (>1 mm, e.g., aggregate culture devices [59]). In the vast majority of microfabrication pipelines, there is a practical gap at the interface between the cellular and multicellular scale, as conventional photolithography is mostly suited for precise features on the smaller scales and is less suited for multicellular scales. Moreover, combining efficiently multiple fabrication rounds across scales is sometimes challenging, time consuming, and error prone. Instead, methodologies such as SOL3D are capable of simultaneously combining cellular, multicellular, and tissue scale features within the same fabrication round, effectively filling a gap in the potential toolkits currently available.

Another advantage here is that SOL3D is not tied to any particular printer or resin, and as such can work for any combination (we have tested several resins over 3 different printers) and with the advent of higher resolution printers, this gap will become significantly smaller. We have also demonstrated the versatility of our pipeline by successfully developing cell culture devices of different designs targeted to a wide range of applications, which either confers new capabilities to conventional culture systems (e.g., easily plating multiple cell types on precise spatiotemporal relationships) or customisation of bioengineered culture systems.

While a number of studies have already proposed similar protocols (e.g., heat curing [60], UV light [61], micro-diamond coating [62]), they all generally tend to either have a considerably lower resolution or in some cases require obligatory steps with very expensive specialised equipment that is not generally available to most biological labs. Moreover, the chemicals and materials used in several of these processes require a much higher degree of training and expertise compared to the pipeline presented here, which requires no hazardous processes or chemicals, but only nontoxic and easily handleable components, and could therefore be implemented in a lab without risk.

In its most minimal implementation, SOL3D requires only a high-resolution (approximately 50 μm) desktop vat polymerisation printer, a suitable resin, PDMS and everyday cell culture, and microscopy components. All of these resources can be obtained with an initial investment below the cost of 2 vials of monoclonal antibodies and at an estimated running cost of <300 USD per year to produce regular medium size batches of devices, which would be affordable to any lab that has an ongoing budget for cell culture consumables. For context, a commercially available single-use 8-well chamber slide suitable for live microscopy with fixed dimensions and no possibility of customisation, ranges between 10 and 15 USD, while a single microfluidic device costs from 50 to 120 USD, both of which can be easily replaced by SOL3D. Moreover, SOL3D resin moulds, like conventional silicon wafer PDMS masters, can be reused efficiently several times with minimal changes in overall fidelity of the replica (S9 Fig), while offering a wider possible range of features at different scales within the same devices. In this use case, repeated reuse (above 20/30 times) might create warping within the mould; however, as the resin moulds are comparatively cheaper and faster to produce, they still offer a potentially more sustainable path to large scale/long-term experiments.

One caveat to the capability of the method, which is a limitation due to the properties of PDMS rather than the fabrication process, is that while it is possible to create complex devices

without any specialised equipment to effectively use smaller features (**Fig 2**), a step of oxygen plasma treatment was necessary. However, plasma etchers are not available to all labs, potentially limiting the applicability of this protocol. Wang and colleagues tested alternatives to plasma treatment of PDMS persorption of fibronectin sufficient for Caco-2 cells [63]; hence, plasma treatment might not be necessary for all cell types and has to be assessed individually.

Unfortunately, the low compatibility of most UV resins with cell culture experiments also poses limitations to the potential application of this method and vat polymerisation in general. We have characterised several different commercially available resins and found that all of them, in a treated or untreated state, are cytotoxic (**S2 Fig**). There are some biocompatible alternatives available on the market (although the extent to which biocompatibility, as defined for dental implants, can be directly applied to stem cell cultures and other more sensitive biological systems needs to be verified) to overcome this fundamental limitation of 3D vat polymerisation, but for a far higher price than any standard resin. Additionally, the fact that resin composition is generally proprietary limits the possible customisations that end-users can achieve. Some providers have recently started publishing their resin composition as well as customised resins generated from a number of research groups that can be recreated, although the burden of adoption of these unique formulations is likely to prevent their use in most cases [64–66]. Alternatively, modifications of PDMS to be printable can lead to decreased resolution of the print, undermining the strong advantage of high-resolution printers for microfabrication using resins [67]. The solution we implemented for the well-known curing inhibition issues with resin moulds and PDMS is to extract and cure completely the resin first with a multistep preparation (see **Fig 1**) and then further shield the PDMS with a layer of enamel coating.

The use of PDMS in cell culture experiments has a long history and we have decades of experience with its use with cells and in microscopy applications; however, it also has some potential drawbacks. In particular, several groups have reported that PDMS can retain organic molecules and adsorb other substances, which can severely impact biological and biochemical experiments in particular cases [32,68–70]. However, one advantage of this setup is that it can be used for any soft-lithography material, for example, polymers such as Flexdym, polymethylmetacrylate, or poly(DL-lactide-co-glycolide), as well as with any hydrogel or cure-forming material with a temperature below 70 degrees.

With this system, we aim to empower any lab, regardless of its capabilities, access to resources or prior expertise, to create customised microdevices with features tailored to their specific biological experiments and designed with their biological question in mind, which will significantly lower the barrier for experimentation with microfabrication and tissue engineering applications in any field.

## 4. Materials and methods

### Cell culture

Control hiPSC motor neuron and cortical neuron progenitors were derived as described [71] from multiple donors (Table 2). These cells were cultured on Matrigel (Corning) coated plates in base medium, comprised of 50% NeuroBasal (Gibco), 50% advanced DMEM (Gibco), supplemented with B27 and N2 (gibco), 100 μg/ml Pen-Strep (Gibco), and 2 mM L-alanyl-L-glutamine dipeptide (Gibco). For expansion of progenitors, FGF (20 ng/ml) (Gibco) was added to base medium. Differentiation of MN progenitors was achieved using base medium with compound E (Enzo) (0.1 μm) and the growth factors BDNF (10 ng/ml) and GDNF (10 ng/ml) unless stated. Astrocytes were generated from iPSC using a modified protocol described in Hall and colleagues [71]. Derived astrocyte progenitors were cultured in neuronal base medium with FGF (20 ng/ml) (Gibco) and EGF (20 ng/ml) (Thermo). For differentiation,

**Table 2. Cell lines.**

| Cell lines | Name | Donor age | Donor sex | Mutations present |
|---|---|---|---|---|
| iPSC/Motor neuron/cortical/astrocyte progenitors | Control 1 | 78 | Male | None |
| iPSC/Motor neuron/cortical/astrocyte progenitors | Control 2 | 64 | Male | None |
| iPSC/Motor neuron/cortical/astrocyte progenitors | Control 3 | Unknown | Female | None |
| iPSC/Motor neuron/cortical/astrocyte progenitors | Control 5 | 51 | Male | None |
| AB1190 Human immortalised myoblasts | Paravertebral muscle | 16 | Male | None |

astrocytes were cultured in base medium without growth factor supplements. All cells were cultured with 5% CO2 in humidified atmosphere at 37˚C (Table 3).

## Transfection of MNPs

For transfection of cells, a plasmid-based *Piggybac* transposon system was used with pgK-Puro-CMV-GFP and pPb-CAG-RFP-Hygro construct cloned in the lab and a *PiggyBac* vector containing the transposase. MNPs were sparsely cultured on a 24-well plate and 1 day after passaging transfected using Mirus LT1 (Mirus Bio) transfection reagent. Plasmids were

**Table 3. Cell culture.**

| Supplier | Reagent/apparatus | Catalogue number | Stock concentration/mass |
|---|---|---|---|
| Gibco | B27 supplement | 17504044 | 50x |
| Gibco | N2 supplement | 17502048 | 100x |
| Gibco | Pen/Strep | 15070063 | 5,000 U/ml |
| Gibco | Neurobasal medium | 21103049 | N/A |
| Gibco | Advanced DMEM/F-12 | 12634028 | N/A |
| Corning | Matrigel | 354230 | N/A |
| Stem Cell Technologies | Accutase | A6964-100ML | N/A |
| Gibco | GlutaMAX Supplement | 35050061 | N/A |
| Millipore | Compound E | ALX-270-415-C250 | 250 µg/ml |
| Invitrogen | EDTA | 15575–038 | 0.5M |
| Gibco | DPBS | 14190–094 | N/A |
| Gibco | FGF | PHG6015 | 50 µg/ml |
| Thermo Fisher | EGF | PHG0311 | 50 µg/ml |
| Beckman Coulter | Allegra 21R centrifuge | BE-A21R | N/A |
| PHCHD | $CO_2$ Incubator | MCO-230AICUV | N/A |
| Thermo Fisher | BDNF | PHC7074 | 10 µg/ml |
| Thermo Fisher | GDNF | PHC7045 | 10 µg/ml |
| Baxter | Thrombin | TISSEELDUO 500 | 3 U/ml |
| Baxter | Fibrinogen | TISSEELDUO 500 | 3.5 mg/ml |
| Thermo Fisher | UltraPure Agarose | 16500500 | 2% |
| Cell guidance systems | ROCK inhibitor (Y-27632) | SM02-5 | 10 µm |
| Promocell | Skeletal muscle growth medium | C-23060 | N/A |
| Sigma | aprotinin | A3428 | 33 µg/µl |
| Sigma | Insulin | 10516 | 0.01 mg/ml |
| Sigma | DMEM | D5671 | N/A |
| Thermo Fisher | E8 flex | A2858501 | N/A |
| Mirus Bio | Mirus LT1 reagent | MIR 2300 | See description |
| Sigma | BSA | A2153-10G | N/A |

added at total 0.5 µg per well (GFP/RFP+ transposase containing plasmid) in 200 µl of Pen-Strep free growth medium. This solution was gently mixed before addition to the wells that also contained 200 µl of Pen-Strep free medium. The medium containing the transfection reagent was exchanged with growth medium after 24 h. Cells were cultured to confluency and then pooled into a 6-well plate for further expansion.

## Fabrication

**3D printing.**  3D-printed moulds were designed using fusion 360 [72] and tinkercad [73] computer-aided design software, then exported as.stl (stereolithography) files to either Chitubox or Photon workshop slicing software. These software were used to define print parameters such as layer thickness, layer UV exposure time, and lifting/retract speeds for each resin. All resins, printers, and print settings can be found in Tables 1 and 4. For an in-depth detailed guide on how to implement the printing pipeline, see Supporting information (**S1 Text**). The designs of all the devices described in this paper are available to download and modify on our lab GitHub page (https://github.com/SerioLab/SOL3D).

## Post processing

After printing, constructs were washed in fresh isopropanol (IPA) using either/or/both an ultrasonic cleaner and stirring washing bath (Anycubic). Washing method and time were varied as part of the protocol establishment. To ensure fair comparison, washing IPA was filtered for every resin for a given washing condition to remove resin components from previous washes. After washing, all prints were cured in a commercially available curing chamber (Anycubic) for 60 min. Constructs were then selectively coated with a layer of enamel paint (Plasti-kote) using a hobbyist airbrushing system (Timbertech) diluted 70:30 with water as per manufacture instructions. Painted casts were left to dry at room temperature on the bench for at least an hour before PDMS casting (Table 5).

**Table 4. 3D printer fabrication.**

| Supplier | Resin | Printer | Layer thickness (µm) | Layer exposure time (s) | Bottom Exposure time (s) | Bottom layers | Transition layers | Light off delay (s) | Z Lift distance (mm) | Lift speed (mm/min) | Retract speed (mm/min) |
|---|---|---|---|---|---|---|---|---|---|---|---|
| Phrozen | Aqua Gray 4K | Phrozen sonic mini 4K | 50 | 6.0 | 50 | 6 | 0 | 9 | 5 | 65 | 150 |
| Phrozen | Water-washable model Gray | Phrozen sonic mini 4K | 50 | 2.5 | 25 | 4 | 0 | 9 | 5 | 65 | 150 |
| Elegoo | ABS-like | Anycubic Photon S | 50 | 9.0 | 60 | 6 | 0 | 1.5 | 6 | 180 | 180 |
| Anycubic | Clear | Anycubic Photon S | 50 | 10.0 | 50 | 8 | 0 | 1 | 6 | 180 | 180 |
| Liqcreate | Premium Toµgh | Phrozen sonic mini 4K | 50 | 6.0 | 75 | 1 | 6 | 9 | 5 | 65 | 150 |
| Liqcreate | Flexible X | Phrozen sonic mini 4K | 50 | 18.0 | 70 | 1 | 6 | 9 | 5 | 65 | 150 |
| Phrozen | Aqua Gray 4K | Phrozen sonic mini 4K | 10 | 1.8 | 37.5 | 6 | 0 | 10 | 5 | 55 | 150 |
| Next Dent | Ortho Clear | Phrozen sonic mini 4K | 50 | 4.6 | 30 | 6 | 6 | 10 | 6 | 60 | 150 |

**Table 5. Post processing.**

| Supplier | Apparatus |
| --- | --- |
| Fisher Scientific | Molecular biology grade isopropanol |
| Anycubic | Wash and Cure 2.0 |
| Life Basis | Ultrasonic cleaner 600 ml |
| KNF | Laboport N86 mini diaphragm vacuum pump |
| Timbertech | Timbertech airbrush ABPST01 with air compressor |
| Plastikote | B35 chrome fast dry enamel paint |
| Thermo Fisher | HeraTherm oven |

## Microfabrication of patterned substrates using soft lithography

Microgroove substrates were manufactured from silicon masters patterned using photolithography as previously described [74]. Briefly, SU-8 2002 (Kayaku) was spun on a silicon wafer for 40 s at 1,000 rpm on a spin coater (Polos) and prebaked at 95°C for 2 min. A microgroove pattern designed in CleWin5 and containing $10 \times 10$ μm grooves with 250 μm plateaus every 5 mm was then etched into the SU-8 via UV exposure and an aligned photomask with the design (Kiss MA6 mask aligner). Excess SU-8 was cleaned with PEGMA, then soft and hard baked at 95°C for 5 min, before being silanised with trichlorosilane ($C_8H_4Cl_3F_{13}Si$) in a vacuum chamber for 1 h. Excess silane was then washed off from masters with 100% acetone. An unpatterned silicon wafer was used for flat substrates. For comparison of surface roughness between 3D-printed casts and microfabricated substrates, etching was achieved by a single photolithographic step using a MicroWriter ML3 (Durham Magneto Optics) to form a pattern designed to mimic the potential capabilities of 3D printing in microfabrication. Following the photolithographic step, PDMS casts (prepared as above) were made of the flat or micropatterned silicon wafers, spin coated at 300 rpm for 40 s on a spin coater (Polos) to ensure uniform thickness and cured on a hotplate at 100°C for 5 to 10 min (Table 6).

## PDMS

Sylgard-184 silicone elastomer kit PDMS pre-polymer was well mixed (5 min) with curing agent at a 10:1 w/w ratio using a digital balance (Sartorius BP610) prior to vacuum desiccation and casting at various temperatures (60 to 90°C).

## Biofunctionalisation

Biofunctionalisation of PDMS substrates and casts from 3D-printed moulds was achieved using oxygen plasma treatment (30 s, 50%, 7sccm–unless stated otherwise) (Henniker Plasma).

**Table 6. Microfabrication.**

| Supplier | Reagent/apparatus | Catalogue code |
| --- | --- | --- |
| Kayaku | SU8–2002 | N/A |
| Karl Suss | MA6 mask aligner | N/A |
| Sigma Aldrich | Trichlorosilane | 448931 |
| Sigma Aldrich | PEGMA | 409537 |
| Fisher Scientific | Acetone | 13277983 |
| Fisher Scientific | Vacuum Desiccator | 11852732 |
| Durham Magneto Optics | MicroWriter ML3 | N/A |
| Polos | 200 NPP Spincoater | 42839 |

**Table 7. PDMS soft lithography and biofunctionalisation.**

| Supplier | Reagent/apparatus | Catalogue code |
|---|---|---|
| Dow Chemical | Sylgard 184 PDMS kit | 01673921 |
| Sartorius | BP610 balance | Z266906 |
| Henniker Plasma | HPT-100 | N/A |
| Gibco | Poly-D-lysine | A38904-01 |
| Sigma Aldrich | Laminin | L2020-1MG |
| Analytik Jena | UV lamp | UVP XX-15S |
| Frenshion | Clear UV Resin | B0823HDLM8 |
| Alonefire | SV003 10W 365nm UV torch | B07SWW5FHB |

Additional biofunctionalisation of micropatterned PDMS substrates to facilitate cellular attachment was achieved using a coat of poly-D-Lysine 0.01% (PDL) (Gibco) for 15 min and laminin (Sigma) overnight (unless stated otherwise) (Table 7).

## PDMS swelling experiments

PDMS samples cured on our Sol3D moulds (approximately 2 g) were pre-weighed and then incubated in *n*-hexane (10 ml) in a sealed glass vial for 24 h at room temperature. After this time, the sample was removed, dried under air, and the mass after swelling recorded again. The *n*-hexane supernatant was evaporated in vacuo and the mass of residue also recorded. The experiment was performed in triplicate, and errors presented as ± standard deviation. Change in mass of samples: −0.0076 ± 0.0005 g; Leached residue: +0.0076 ± 0.0005 g.

## Cell seeding

Cells were detached from culture using Accutase (Stem Cell Technologies) and seeded in oxygen plasma-treated PDMS constructs cast from 3D-printed moulds bound to either tissue culture plastic, flat PDMS, or micropatterned PDMS substrates. Cells were concentrated to 300 μl per detached well and seeded in differentiation media with Compound E (Milipore). Following initial plating, cells were left to settle for 2 h in constructs. Cells were then washed 2× with PBS to remove unattached cells from constructs and wells filled with differentiation media supplemented with Compound E with constructs left in place. After 24 h in culture, cells were washed with PBS and constructs removed before a 1:100 Matrigel spike for >2 h and culturing cells (as above) in differentiation media supplemented with Compound E for 7 days (or as stated). For longer term experiments, media was selectively supplemented with additional BDNF and GDNF growth factors depending on the experiment.

## Immunostaining

Prior to staining cells were fixed in 4% PFA (Boster) for 15 min (unless stated otherwise) and washed 3× with PBS. Cells were then permeabilised with 0.1% Triton-X for 10 min and blocked with 3% goat serum (GS) (Sigma) for 30 min at room temperature (unless stated otherwise). Antibodies diluted in 0.05% Triton-X and 1.5% GS in PBS were then used to stain cells for markers of interest. Antibodies and their concentrations can be found in Table 8. Primary antibodies were incubated for 1 h at room temperature in the dark, before washing 3× with PBS. Secondary antibodies were then incubated for 30 min at room temperature in the dark, before washing 3× with PBS. All secondary antibodies were incubated at a final concentration of 2 μg/ml (Table 9). For some experiments, stained cells were then mounted on glass slides using FluorSave (Millipore). Otherwise, cells were kept in PBS at 4°C until imaging. For

**Table 8. Primary antibodies/stains.**

| Supplier | Antigen | Species | Isotype | Clone | Catalogue code | Dilution |
|----------|---------|---------|---------|-------|----------------|----------|
| Sigma Aldrich | βIII-Tubulin | Mouse | IgG2b | SDL.3D10 | T5076 | 1:1,000 |
| Thermo Fisher | MAP2 | Mouse | IgG1 | M13 | 13–1,500 | 1:500 |
| Abcam | GFAP | Chicken | Polyclonal | Polyclonal | ab4674 | 1:500 |
| Thermo Fisher | DAPI | N/A | N/A | N/A | 62248 | 1:5,000 |
| Spirochrome | Silicone Rhodamine-tubulin kit | N/A | N/A | N/A | SC002 | 1:10,000 |
| DSHB | MF-20 | Mouse | IgG2b | MYH1E | MF-20 | 1:9 |
| DSHB | Titin | Mouse | IgM | TTN | 9 D10 | 1:20 |

live cell imaging, cells were incubated with 1:10,000 (100 nM) silicon rhodamine tubulin (SiR) live cell dye (Spirochrome) for 1 h before removal of the dye and imaging (Table 10).

## Microscopy

Cells were imaged using an encased Nikon eclipse TE2000-E fluorescence microscope running Micromanager software with 4×, 20× LWD and 20× SWD objectives, cool LED pE-4000 16 LED light source, and a Prior controlled stage. An LED driver Arduino controlled light source provided illumination for brightfield imaging. To allow longitudinal and live imaging, the microscope chamber was humidified and heated to 37.0˚C with 5% $CO_2$ using a CAL3300 incubator temperature regulator (Solent Scientific) and $CO_2$ regulator (Okolab). Humidity, $CO_2$ balance, and temperature were regulated by further encasing the plate in a sealed custom 3D-printed chamber with humidified $CO_2$ inlet (Table 11).

## Surface characterisation

**Optical profilometry.** Quantification of 3D-printed mould dimensions, surface roughness, and pixel size was achieved using a Sensofar S Neox optical profilometer to measure features in X and Y, and layer thickness. Multi-image z-stacks were captured over stitching areas with 25% overlap using a 20× Nikon EPI objective and surface-variation scanning mode. For analysis of patterned silicon master feature dimensions and PDMS cast surface roughness from both 3D-printed moulds and silicon masters, multi-image z-stacks were captured of stitching areas with 25% overlap using a 20× Nikon DI objective and confocal scanning mode. Analysis of features was conducted using in-built analysis tools. Plane correction was conducted on all images to reduce bias within imaged ROIs.

**SEM.** Samples of coated and uncoated 3D-printed moulds were sputter coated with a 10-nm thick layer of Platinum using a Quorum Q150R coater and imaged by a Phenom ProX Desktop SEM (Thermo Scientific) at an acceleration voltage of 10 kV (unless noise was too high, then 5 kV was used). Images of the surface and cross section of prints was captured to

**Table 9. Secondary antibodies.**

| Supplier | Host species | Target species | Isotype | Conjugate | Catalogue code | Dilution |
|----------|--------------|----------------|---------|-----------|----------------|----------|
| Invitrogen | Goat | Mouse | IgG2b | Alexa Fluor 647 | A-21242 | 1:1,000 |
| Invitrogen | Goat | Mouse | IgG1 | Alexa Fluor 555 | A-21127 | 1:1,000 |
| Invitrogen | Goat | Chicken | Polyclonal | Alexa Fluor 555 (H+L) | A-21437 | 1:1,000 |
| Invitrogen | Goat | Mouse | IgG2b | Alexa Fluor 546 | A-21143 | 1:1,000 |
| Invitrogen | Goat | Mouse | Polyclonal | Alexa Fluor 488 | A-21042 | 1:1,000 |

**Table 10. Immunocytochemistry.**

| Supplier | Reagent | Catalogue code |
|---|---|---|
| Sigma Aldrich | Goat serum | G9023-10ML |
| Invitrogen | 1% Triton in PBS | HFH10 |
| Millipore | Fluor Save | 345789–20 ml |
| Fisher Scientific | 1 mm microscope slides | 15545650 |
| Fisher Scientific | 22 × 22 glass coverslips | 12312128 |
| Fisher Scientific | 60 × 24 glass coverslips | 10083957 |
| Boster | 4% Paraformaldehyde | AR1068 (500 ml) |
| Sigma Aldrich | Brand Cavity Slides | BR475535-50EA |
| Thermo Fisher | Fluoromount G | 00-4958-02 |

investigate the thickness of applied paint and identify changes in surface roughness/topography. Images were processed in ImageJ FIJI [75] (Table 12).

## Image analysis

All image analysis was conducted in ImageJ FIJI software, processed using R, and presented with R or super-plots-of-data app [76] unless stated otherwise.

**Line graph analysis.** To quantify cellular segregation within the same device/multiple devices threshold fluorescence intensity was adapted to improve signal-to-noise ratio. A rectangular area was then drawn over the cells of interest and fluorescence intensity plots were obtained for each point.

**Area and aspect ratio.** Measurements of seeded cell area and aspect ratios were compared to either CAD specifications or 3D-printed mould feature dimensions. Cell measurements were taken from the borders of aggregates using tubulin markers Silicon-Rhodamine tubulin (live) and βIII-tubulin (fixed). 3D-printed mould dimensions were obtained using SensoScan software in-built analysis tools.

## Resin biocompatibility

Chips from each of the 6 resins printed using 50 μm layer thickness printer settings (Table 4) and post processed as above (20 min sonication and wash, 60 min UV cure) without enamel coating. Chips were then either sterilised with UV for 15 min, or baked for 4 h at 75°C, washed in PBS for 72 h at 50°C, and UV sterilised for 15 min before being added to cultures of MNPs pre-stained with SiR-tubulin cytoskeletal live dye. Cells incubated with untreated chips were left for 48 h before imaging. Cells incubated with treated resin chips were live imaged for the first 24 h in culture, then again at 48 h with control wells not containing any resin. Investigation of the Ortho dental clear resin was performed on cylinder-shaped resin prints that can be used as an inset to tissue culture wells. The resin prints were either treated using our SOL3D

**Table 11. Microscope hardware.**

| Supplier | Apparatus | Catalogue code |
|---|---|---|
| Nikon | TE-2000-E | N/A |
| Solent Scientific | CAL3300 incubator temperature regulator | N/A |
| Cool LED | pE-4000 16 LED light source | N/A |
| Okolab | $CO_2$ regulator | N/A |
| Zeiss | LSM710 Inverted Confocal microscope | N/A |

**Table 12. Surface characterisation.**

| Supplier | Reagent/apparatus | Catalogue code |
|---|---|---|
| Phenom | ProX Desktop SEM | N/A |
| Sensofar | S Neox optical profiler | N/A |
| Quorum | Q150R coater | N/A |

protocol or the washing protocol suggested by the manufacturer. The processed resin tubes were then inserted into wells containing a known number of iPSC-derived MNs and cocultured for 4 days.

## Microwell arrays

Microwell arrays of repeating 400 μm × 400 μm × 150 μm pyramids were 3D printed at 10 μm layer thickness (Table 4) and post processed as above (20 min sonication and wash, 60 min UV cure) without enamel coating. PDMS (prepared as above) was casted and left un-biofunctionalised to enhance EB formation through PDMS hydrophobicity. Single IPSCs pre-stained with SiR-Tubulin live dye were then seeded on microwell arrays or flat PDMS in E8 flex medium (Thermo) and centrifuged for 1 min at 100 rpm to settle cells into microwells. Cells were cultured for 4 days, imaged directly after seeding, and at days 2 and 4 of culture. After imaging on day 4, the regular EBs were detached from microwells and imaged to measure their size.

## Chamber slide

**Manufacture.** A 3D design with 12 wells with 5 mm diameter funnel and 3 mm diameter 1 mm deep straight well was 3D printed with Resin A at 50 μm layer thickness (Table 4) on the Phrozen Sonic mini 4K and post processed as above (20 min sonication and wash, 60 min UV cure) with enamel coating. PDMS (prepared as above) casts were made of the arrays and bonded to glass coverslips with 2 methods.

**PDMS bonding.** *Oxygen Plasma*. The surface of PDMS casts from 3D-printed moulds and glass coverslips were oxygen plasma treated (30 s, 50%, 7sccm) (Henniker Plasma), sealed together, and baked at 75˚C for 15 min.

*UV glue*. The surface of PDMS casts from 3D-printed moulds was sealed on glass coverslips and clear photopolymerisable resin was applied round the exterior of the PDMS. Through visual inspection, we verified that the glue is only applied on the outside of the device and cannot physically leak into the wells. This was verified though live cell imaging on the chamber device, a UV resin is highly toxic for cells and would have resulted in immediate cell death. Resin was cured via 1-min UV exposure with a 365 nm UV torch (Alonefire).

*Biofunctionalisation*. Plasma and UV bonded PDMS chamber slide devices were then sterilised with UV for 15 min (Analytik Jena UV light) and biofunctionalised with a coat of poly-D-Lysine 0.01% (PDL) (Gibco) for 15 min and laminin (Sigma) overnight in each well.

*Seeding*. Control 3 astrocyte progenitors pre-stained with SiR-tubulin live dye were then seeded (as defined in methods section–Cell seeding) after concentration (1 confluent well in 400 μl media) in expansion media with decreasing density in alternate wells of the chamber slide device. Cells were seeded in 50 μl, 25 μl, 12.5 μl, 6.25 μl fractions of the concentrated cell stock, left to settle for 10 min, before additional media was added. Cells were cultured as before (Methods–Cell culture) and imaged after 24 h to qualitatively assess viability at different densities.

*Microfluidics*. Designs were prepared in Fusion360 and printed in 8k resin using the Phrozen 8k Mini printer. After the print was processed using the SOL3D protocol, with washing and curing. PDMS was cast on the print and peeled off. Then, holes with a diameter of 1.5 mm were punched into the respective positions. A glass slide and the microfluidic device were oxygen plasma treated for 30 s with 50% power, the immediately pressed together for permanent bonding. For final bonding, the assembled device was heated to 100˚ for 20 min. After cooling, a tubing with 1.6 mm diameter was inserted into the punched wells. Then, a fluorescein solution was sent through the microfluidic device using a syringe, while imaging on an EVOs microscope.

## 3D engineered muscle

**Preparation, differentiation, and culture.** 3D muscle constructs were prepared as previously described [50,52]. Rock inhibitor (Cell guidance systems) at a concentration of 10 μm was added to cells 2 h prior to preparing gels, and $10^6$ human immortalised myoblasts (line AB1190, kindly supplied by the Myoline Platform of the Institute of Myology, Paris, France) cells were used in a total volume of 120 μl comprised of 3.5 mg/ml of human fibrinogen (Baxter, TISSEELDUO 500), 3 U/ml of thrombin (Baxter, TISSEELDUO 500), 10% Matrigel (Corning, 356231), and inactivated myoblast medium (20 to 30 min at 56˚C) (PromoCell skeletal muscle cell growth medium, C-23060). The mix was pipetted into agarose moulds containing posts and placed at 37˚C, 5% $CO_2$ for 2 h allowing hydrogels to polymerise. These moulds were prepared in 12-well plates using 2% UltraPure agarose (Thermo Fisher Scientific, 16500500). A ring was placed underneath the arms of the posts and inserted onto a 12-well plate to determine the distance the posts can be pushed down into the agarose. DMEM (Sigma, D5671) was then added to the construct, and the posts containing the muscle construct was removed from the agarose mould and placed into a new 12-well plate with myoblast medium containing 33 μg/μl aprotinin (Sigma, A3428) at 37˚C, 5% $CO_2$. After 48 h, the muscle construct was placed in differentiation media (DMEM with 0.01 mg/ml insulin (Sigma, 10516) and 33 μg/μl aprotinin); media was changed every other day. Work with human cells was performed under approval of the NHS Health Research Authority Research Ethics Committee (reference no. 13/LO/1826) and Integrated Research Application System (IRAS) project (ID No. 141100).

**Contractility analysis.** Contractility of engineered muscles [77] was achieved with a pair of autoclaved pacing carbon electrodes (EHT-technology) mounted in the well containing the 3D muscles dipped gently into the media. The stimulator was set to deliver 5 ms bipolar square pulses of 20 mV with 0.5 Hz frequency. Muscle contraction and post holder movements were recorded over a period of 5 s during stimulation via DinoLite Edge microscope (DinoLite) mounted underneath the posts of each device. Analysis of footage was conducted in Imagej using the MuscleMotion plugin (Table 13).

**Immunocytochemistry.** After contractility recordings, muscle constructs were fixed in 1% PFA overnight before removal from posts. Constructs were then permeabilised and blocked for 6 h in TBS 0.05 M (1×) pH 7.4, 10% FBS, 1% BSA, 0.5% Triton X-100 at 4˚C. Prior

**Table 13. Microstimulation hardware.**

| Supplier | Reagent/apparatus | Catalogue code |
|---|---|---|
| EHT Technologies | Silicone posts | C0001 |
| EHT Technologies | Carbon pacing electrodes | P0001 |
| DinoLite | USB microscope | AM73915MZT |

to overnight incubation with primary antibodies for Titin and MyHC (Table 8) at 4˚C in TBS 0.05 M (1×) pH 7.4, 1% BSA, 0.5% Triton X-100. The following day, constructs were washed 6 times in PBS before overnight incubation with secondary antibodies and DAPI at 4˚C in TBS 0.05 M (1×) pH 7.4, 1% BSA, 0.5% Triton X-100. Finally, constructs were mounted on Brand Cavity Slides with Fluoromount G (Thermo) prior to imaging.

**Differentiation and directionality analysis.** Images of 3D muscle constructs were captured using an inverted Zeiss confocal with a 40× objective. Z-stacks were taken across the constructs and projected using the SUM function in ImageJ. Single images were isolated and directionality measured from titin signal using the directionality plugin in ImageJ.

## Rheometric analysis

PDMS 1:10 and 1:20 rheological properties were measured after 60 min of curing at 75˚C, by means of the Discovery HR20 rheometer (TA instrument). Samples were analysed at room temperature (25˚C), using an 8 mm plate geometry, with a frequency sweep of 1 and an angular frequency from 25.12–0.1 rad/s. Storage (G') and loss (G") moduli were measured for the entire period. Young modulus (E) was calculated following the equation [78].

$$E = 2G'(1 + v)$$

Where $G'$ is the storage modulus and $v$ is the Poisson ratio (0.5).

## Hydrogel culture system

**Hydrogel composition.** Preparation steps started by measuring neuronal culture medium (4 ml) and homogenously mixed with Matrigel (Corning) (1 ml). Hyaluronic acid (5 mg/ml) from Sigma Aldrich was mixed in the above solvent and stirred for overnight (12 h) at ambient temperature. This was followed by addition of Fibrinogen (45 mg/ml) from Sigma Aldrich and stirred for 5 h at room temperature. As a final step, Alginate (5% (w/v)) from Sigma Aldrich is added to the above mixture and was allowed to be stirred overnight to obtain a homogenous hydrogel matrix. MNPs were dissociated from a 6-well plate using EDTA in PBS and were gently mixed with the hydrogel matrix (4 to 5 million cells/ml) by pipetting them up-down gently for homogenous distribution of cells throughout the hydrogel, this led to the formation of bio-ink. This bio-ink is crosslinked by a 50:50 solution consisting of calcium chloride ($CaCl_2$) (1.5% (w/v)) and thrombin (25 U/ml in 0.1% BSA Solution). Bio-ink is allowed to be crosslinked for the duration of 15 min at room temperature. After crosslinking, the bio-ink was washed with PBS solution for 3 times. This was followed by flooding the wells gently through the walls of cell plate with neuronal culture media supplemented with Compound E and kept in incubator at 37˚C with 5% $CO_2$.

## Supporting information

**S1 Fig. Comparison of costs for 3D vat polymerisation printing and photolithography with all the required equipment and materials but excluding personnel training costs.** (DOCX)

**S2 Fig. SLA resins themselves are toxic with and without pretreatment.** (A) Chips of 6 resins were added to cultures of motor neuron progenitors pre-stained with SiR–tubulin live dye, incubated for 48 h before imaging. Representative images of SiR live dye-stained motor neurons after 48 h in culture with resins compared to 2 control wells. (B) Chips of 6 resins were treated with extra processing steps to improve biocompatibility, bake 4 h at 75˚C, wash in PBS 72 h at 50˚C, UV sterilise 15 min before being added to cultures of motor neuron progenitors

pre-stained with SiR-tubulin cytoskeletal live dye, incubated for 48 h before imaging. Representative images of SiR live dye-stained motor neurons after 48 h in culture with treated resins compared to 2 control wells. (C) Representative time-lapse videos of SiR live dye-stained motor neurons during the first 24 h in culture with treated resin chips compared to 2 control wells.
(DOCX)

**S3 Fig. Effects of coculture with a biocompatible resin on iPSC-derived MNs.** iPSC-derived MNs were plated at equal densities. Then, a 3D-printed well-sized cylinder of ortho-clear resin was added to the well and incubated for 4 days to identify the toxic effects of the biocompatible resin. The resin was either pretreated with the recommended manufacturer's protocol or our pipeline. Cells were stained for β-3-tubulin and DAPI.
(DOCX)

**S4 Fig. Quantitation of PDMS curing on 3D-printed moulds.** (A) Dendrogram of the spectral similarity of PDMS casts from 3D-printed moulds fabricated with 6 commercially available resins, washed with 5 conditions (S + W = sonicate 10 min, wash 10 min, S10 = Sonicate 10 min, S20 = sonicate 20 min, W10 = Wash 10 min, W20 = Wash 20 min), either untreated or coated with airbrush (AB) and cured at 3 different temperatures (60, 75, and 90), compared to samples of uncured and cured PDMS. Replicate and print number (RxPx) for each condition. (B) Heterogeneity of PDMS cast curing from 3D-printed moulds by observation ranging from "cured," "partially cured," "not cured," "damaged."
(DOCX)

**S5 Fig. MALDI-TOF analysis of leachates from SOL3D PDMS devices.** Spectra of samples taken after incubating PDMS devices fabricated in SOL3D moulds in water at 37˚C for 72 h, to mimic the conditions of the biological experiments, compared to a control consisting of water maintained at 37˚C for 72 h within the same incubator and the technical control for the analysis employing only MALDI matrix. The comparison shows no detectable leachates in water incubated with the SOL3D PDMS devices, with a profile essentially comparable to what observed in the water only control.
(DOCX)

**S6 Fig. Biocompatibility of PDMS cast in 3D-printed moulds with cells (time-lapse and long axons).** (A) Representative β-III Tubulin stained differentiated motor neurons cultured on a flat substrate. (B) Brightfield images of differentiated motor neurons with a PDMS cast from a 3D-printed mould in the culture medium. (C) β-III Tubulin stained motor neuron long axons seeded on PDMS microfabricated substrate. (D) Snapshots from 60-h time-lapse of SiR–tubulin live dye stained motor neuron differentiation on PDMS substrate cast in a 3D-printed mould.
(DOCX)

**S7 Fig. Analysis of paint layer thickness and surface roughness.** (A) Representative SEM image of a resin A print with airbrushed enamel paint. (B) Analysis of paint layer thickness on 3D vat printed moulds. (C–F) Optical profiling of a 3D print (C) painted with enamel, shows a surface roughness that is variable around 3 μm (D) a flat silicon wafer shows variability in the nanometre scale (E, F) a non-painted 3D print showing a small variation based on the pixel size of the screen, with 1 μm variability.
(DOCX)

**S8 Fig. PDMS stencil devices manufactured using photolithography.** (A) Dimensions of the stencil-like device with photolithography. (B) Schematic overview of the seeding strategy with

stencil-like devices. (C) Representative image of a stencil-like device with pockets for cell seeding (arrow). (D) Demonstration of cell seeding using food colouring as a "single-cell suspension." The liquid was manually pipetted on the pockets of the device (arrow). Limited volume can be used. (E) Representative Brightfield image of single motoneurons seeded using a stencil device manufactured using photolithography and a schematic showing the seeding process as well as the position of the cells. (F) Representative Brightfield image of single motoneurons seeded using a plating device manufactured using SOL3D and a schematic showing the seeding process as well as the position of the cells.
(DOCX)

**S9 Fig. The surface of 3D prints is rough.** (A) Representative optical profiles from PDMS casts demoulded from a single airbrushed 3D-printed mould and a single microfabricated mould. (B) Representation of 5 ROI selection for quantification of surface roughness on PDMS casts. (C) Quantitation of surface roughness of PDMS casts from the same device over time (25 simultaneous casts) between an airbrushed 3D-printed mould and a microfabricated mould.
(DOCX)

**S10 Fig. Fluidic seal for cell plating devices.** (A) Schematic (top) and representative image (bottom) of clamping approach to ensure fluid seal when devices are placed on PDMS microgroove substrates. (B) Comparison of liquid seal integrity of clamping strategy (top) compared to open curing (bottom) on PDMS microgroove substrate with different well sizes and shapes using dyed liquid. Successful sealing of devices cast with a glass cover (**Green zoom**). Dye spreads throughout device and grooves using open cured (**Red zoom**). Arrows highlight liquid spreading.
(DOCX)

**S11 Fig. 3D print dimensions are homogenous.** (A) Graph comparing X and Y dimensions of 3D printed constructs to CAD specifications in a single device with well dimensions ranging from 600 μm × 1,000 μm to 50 μm × 1,000 μm. (B) Graph comparing X and Y dimensions of 300 μm × 1,000 μm features on 3D printed constructs to CAD specifications for 6 commercially available resins printed on two 3D printers at manufacturer default settings with a 50 μm layer thickness. (C) Graph comparing the actual layer thickness of 3D printed constructs to CAD specifications for 6 commercially available resins printed on 2 SLA 3D printers at manufacturer default settings with a 50 μm layer thickness. Data points for all graphs can be found in file S11A–S11C Data in **S1 Data**.
(DOCX)

**S12 Fig. Non-plasma devices cannot be used for seeding in microwells.** (A) Representative SiR-Tubulin images of motor neuron progenitors seeded in microwells ranging from 600 μm × 1,000 μm to 50 μm × 1,000 μm. Difference in colour indicates depth in focal plane where cells do not reach the micropatterned substrate below.
(DOCX)

**S13 Fig. Spatiotemporal control over cell seeding in an open well.** (A) Representative SiR tubulin live cell dye fluorescence images of astrocyte and cortical progenitors. Images were captured 2 days following device removal. (B) Channel split of complete circuit after 19 days of culture. GFP transfected motor neurons = green, Glial Fibrillary Acidic Protein (GFAP) identifies astrocytes, β-III–Tubulin identifies cortical neurons and the tubulin in GFP+ motor neurons. Blue device well shapes overlaid for illustrative purposes.
(DOCX)

**S14 Fig. Plating devices enable manual segregated seeding of different cell types in the same well, device or multiple devices.** (A) Multiple PDMS casts from 3D-printed devices can be seeded in the same well and seeded with different cell types at different time points. Schematic overview of the multi-device protocol for seeding GFP and non GFP+ motor neurons at different time points in 3 devices in the same well. (B) Representative fluorescence images of cells GFP+ motor neurons seeded in the inner and outer rings of the 3 devices and imaged with devices still on (left) and after seeding of the second non-GFP+ motor neurons in the medial ring after device stripping (right). Representative line profile of imaged cell fluorescence showing segregation of individual populations to their designated rings.
(DOCX)

**S15 Fig. Plating devices enable geometric manipulation of aggregoid cultures in 2D and 3D.** (A) Schematic overview of protocol for manipulating aggregate geometry in combination with existing microgroove and flat substrates. (B) Representative images of SiR-tubulin live-cell-stained motor neuron aggregoids at day 2 (top) and β-III Tubulin staining after 11 days (bottom) of culture. (C) Boxplot of aggregoid aspect ratio fold change by shape between CAD (blue line) and day 2 of culture from β-III-Tubulin channel (top). (D) Boxplot of aggregoid area fold change by shape between CAD (blue line) and day 2 of culture from β-III Tubulin channel. (E) Representative images of SiR-tubulin stained cortical aggregates after seeding and 24 h later in different geometry stencil devices, round (left), triangular (middle), and rectangular (right).
(DOCX)

**S16 Fig. Representative image of an optical profile from SOLID manufactured triangular grooves.**
(DOCX)

**S17 Fig. 3D-printed mould for microfluidic devices.** (A) Design of a 3 compartment device. (B) Optical profilometer images with indicated positions of measurement (red circle, blue triangle). (C) Profile of dimensions across the channel (blue line in B) with measured depth and width. (D–F) Time-lapse images of a fluorescent solution spreading through the device.
(DOCX)

**S18 Fig. Optimisation and protocol for 3D muscle culture using PDMS constructs from 3D-printed moulds.** (A) Images show 3 different mould sizes tested to optimise the volume of the hydrogel mix at a side and bottom view. (Bi–iii) Images showing moulds with (yellow arrows) and without grooves at various perspectives. (Biv) Image shows top view of agarose once mould is removed. (Ci–ii) Images show posts with and without a ring placed underneath the arms from a front and top view. Rings were designed to match the size of 1 well of a 12-well plate. (Ciii) Images showing posts inserted into agarose moulds at a side view and bottom view from underneath the plate, rings allow posts to be inserted at a specific height.
(DOCX)

**S1 Text. Supplementary guide.**
(DOCX)

**S1 Data. Data points for Figs 3G, 3H, 6E, and S11A–S11C.**
(ZIP)

## Acknowledgments

Human immortalised myoblasts were kindly provided by the Myoline platform of the Institut de Myologie, Paris, France to the Tedesco lab.

## Author Contributions

**Conceptualization:** Cathleen Hagemann, Eugenia Carraro, Christopher D. Spicer, Albane Imbert, Andrea Serio.

**Data curation:** Cathleen Hagemann, Matthew C. D. Bailey, Andrea Serio.

**Formal analysis:** Cathleen Hagemann, Matthew C. D. Bailey, Eugenia Carraro, George Konstantinou, Andrea Serio.

**Funding acquisition:** Andrea Serio.

**Investigation:** Cathleen Hagemann, Matthew C. D. Bailey, Eugenia Carraro, Ksenia S. Stankevich, Valentina Maria Lionello, Noreen Khokhar, Pacharaporn Suklai, Carmen Moreno-Gonzalez, George Konstantinou, Sudeep Joshi, Mads S. Bergholt, Andrea Serio.

**Methodology:** Cathleen Hagemann, Matthew C. D. Bailey, Christina L. Dix, Eleonora Giagnorio, Mads S. Bergholt, Albane Imbert, Francesco Saverio Tedesco, Andrea Serio.

**Project administration:** Cathleen Hagemann, Andrea Serio.

**Resources:** Kelly O'Toole, Mads S. Bergholt, Francesco Saverio Tedesco.

**Supervision:** Andrea Serio.

**Visualization:** Andrea Serio.

**Writing – original draft:** Cathleen Hagemann, Matthew C. D. Bailey, Albane Imbert, Francesco Saverio Tedesco, Andrea Serio.

**Writing – review & editing:** Cathleen Hagemann, Eugenia Carraro, Christopher D. Spicer, Andrea Serio.

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
