## [Editor Report · Decision Letter 0]

18 Jul 2023

Dear Andrea,

Thank you for submitting your manuscript entitled "SOL3D: Soft-lithography on 3D vat polymerised mouldsfor fast, versatile, and accessible high-resolution fabrication of customised multiscale cell culture devices with complex designs" for consideration as a Methods and Resources by PLOS Biology.

Your manuscript has now been evaluated by the PLOS Biology editorial staff and I am writing to let you know that we would like to send your submission out for external peer review. I should say that, unfortunately, we haven't been able to get input from an Academic Editor on your submission. We do not want to hold up the process any further so we will consult with one of our academic editors once the reviewers have submitted their reports. 

Once your full submission is complete, your paper will undergo a series of checks in preparation for peer review. After your manuscript has passed the checks it will be sent out for review. To provide the metadata for your submission, please Login to Editorial Manager (https://www.editorialmanager.com/pbiology) within two working days, i.e. by Jul 20 2023 11:59PM.

Kind regards,

Christian

Christian Schnell, PhD

Senior Editor

PLOS Biology

cschnell@plos.org

---

## [Decision Letter · Decision Letter 1]

5 Sep 2023

Dear Andrea,

Thank you for your patience while your manuscript "SOL3D: Soft-lithography on 3D vat polymerised moulds for fast, versatile, and accessible high-resolution fabrication of customised multiscale cell culture devices with complex designs" was peer-reviewed at PLOS Biology. It has now been evaluated by the PLOS Biology editors, an Academic Editor with relevant expertise, and by several independent reviewers. Apologies again for the long delay, the decision was a bit more challenging than usual as you will see when reading the reports.

In light of the reviews, which you will find at the end of this email, we would like to invite you to revise the work to thoroughly address the reviewers' reports.

As you will see below, the reviewers overall find the manuscript interesting and the method useful. However, they also raise some concerns, in particular about the novelty of the approach and the narrative of the manuscript. To address these issues, after discussion with the Academic Editor, we recommend to improve the narrative of the manuscript to make it useful to a non-expert reader along the lines of the suggestions of Reviewer 2 and Reviewer 3, including the chemical analysis to ensure that the results can be reproduced by the community.

Given the extent of revision needed, we cannot make a decision about publication until we have seen the revised manuscript and your response to the reviewers' comments. Your revised manuscript is likely to be sent for further evaluation by all or a subset of the reviewers.

**IMPORTANT - SUBMITTING YOUR REVISION**

*Re-submission Checklist*

*Published Peer Review*

*PLOS Data Policy*

*Blot and Gel Data Policy*

Sincerely,

Christian

Christian Schnell, PhD

Senior Editor

PLOS Biology

cschnell@plos.org

REVIEWS:

Reviewer #1: In this manuscript, the authors reported their work on using 3D printing and PDMS for customized cell culture devices. The authors examined cytotoxicity, biocompatibility, and cell viability of several 3D printing resins and various structures of PDMS devices fabricated by post-processed 3D printed molds with cells and motor neurons. Their results demonstrate that a simple treatment of 3D-printed molds enables efficient curing of PDMS on the molds, and biofunctionalization of PDMS allows culturing cells in a controlled manner. The manuscript is well-written with various control and sample tests, and the conclusion is supported by carefully presented experimental data. The length, figures, and citations are appropriate, and the results presented in this paper are highly interesting. Therefore, the readership of this work will be broad. Other researchers interested in easy, fast, and cost-effective fabrication of their proof-of-concept culture designs, structures, and experiments without specialized equipment and expertise in microfabrication processes will find this manuscript useful. One of the popular 3D printers providing large scale and high-resolution printing is the FormLabs 3D printer. Have the authors considered testing this printer and their resins? Other than that, I recommend publishing this manuscript as is.

Reviewer #2: PBIOLOGY-D-23-01723 "SOL3D: Soft-lithography on 3D vat polymerised moulds for fast, versatile, and accessible high-resolution fabrication of customized multi scale cell culture devices with complex designs " provides a tutorial-like manuscript guiding researchers that may lack a background in technology but are interested in 3DP custom designs to conduct their cell based research. The timing for this type of paper is excellent, as a growing number of resins is available and the budget printers have decent performance. Reading the article, however, I felt it lacks focus. 

The most significant shortfall for non-technology oriented researchers it its failure to discuss where the designs come from, It does not need to be a full CAD tutorial, but some explanation about where-hoe to draw, file formats, downloadable designs and slicing options are essential to allow new researchers to enter this space. With that, perhaps some hints on hydrodynamic resistance and how design can be used to manipulate flow would aid in the starting up phase. And some information in regard to world to chip interfacing through printing fittings and connectors is also expected to facilitate the journey. 

While the manuscript is lengthy, it lacks focus and a clearly defined scope. In the introduction, a more comprehensive overview of resin printers could be provided. Pleaqse note the nanoscribe and Up3D are commercial 2PP systems and that should be mentioned. Also, BMF has launched the S230, 1 2 um pixel, 10 um resolution printer. For the more budget versions, the difference between DLP where a digital light engine is used, to the LCD based printers where light from an LED array is selectively masked by the screen as well as the SLA based systems where the laser writes the design in a layer by layer manner. Advances have been made in all classes, so a brief summary of what printer provides what edge may help readers with the very first step: buying a printer.

In the experimental, no information is provided on the key difference between two printers and why they use different resins: the wavelength! This is critical information that needs to be understood before people purchase the wrong resin, or want to blend their own. I woudl also suggest to refer to the 2016 (9??)Nordin paper in optimizing exposure, for finer detail and custom resins, this is important. 

The enamel paint is a nice idea, and i would love to understand how it changes morphology through reference SEMs and surface area roughness before coating. It may well be the particles in the paint rather than the 3DP that causes the roughness. The profiler images look nothing like a DPL print, where we expect ~8 um deep pixelation. 

The PDMS casting study is interesting, and needs a reference to Folch' paper on 3DP PDMS. There are many reports casting from DLP moulds, so an expression what is new and what matches precedence is helpful. 

I feel the cell culture part goes in too much detail and would be better have some insight in the breath of the opportunities the work opens up. It is well established that biocompatibility strongly depends on cell type and experiment, and there are some reviews looking at this in detail that could be referenced. 

I would like to suggest to expand the introduction to understand printer, drawing and design and resin selection, while shortening the culture examples to make them nice illustrations of what can be achieved. 

Reviewer #3: Overall assessment and general Comments:

In the paper: "SOL3D: Soft-lithography on 3D vat polymerised moulds for fast, versatile, and

accessible high-resolution fabrication of customised multiscale cell culture devices with

complex designs", Prof. Serio and coworkers present a workflow for casting

silicone (PDMS -Sylgard 184) cell culture parts and devices in molds created using commonly available SLA printers and resins. Compared to traditional soft lithography, the main advantage of this procedure is a somewhat simpler workflow, and the reliance on much cheaper equipment and infrastructure. The main downsides are a worse spatial resolution (~50-100 µm minimal feature sizes in SLA) and some unknowns pertaining to biocompatibility. 

Several other groups have investigated and reported such a workflow. A common challenge is the inhibition of PDMS curing by residual small molecules and oligomers in the resin. Incomplete PDMS curing can have severe impact on biocompatibility, and resin residuals may similarly be cytotoxic, which is combatted through extensive cleaning (washing) and coating of the SLA parts. In light of the earlier work within this space, the novelty of this work is somewhat limited. 

In my view, the main finding of the paper is of practical nature, in the identification of a commercial resin for SLA printing that does not impair PDMS curing notably, and where PDMS parts are fully biocompatible within the tested ranges. However, the reasons why this particular resin is advantageous is not investigated. The paper therefore does not provide any mechanistic insights into the choice or formulation of resin for such purposes. I understand that this can be challenging, since the applied SLA resins are commercial products, but perhaps some chemical analysis would return insights as into why this resin is more compatible than the other tested resins. 

The paper also presents a somewhat workable solution to applying other SLA resins, through coating the printed parts with enamel paints using airbrush. However, this procedure to me seems to not be very reliable for complex e.g. non-planar designs, in particular being a manual process, with the variability this brings. Lastly, even if reproducible, the application of a 30um coating will be problematic for achieving high-resolution features. 

Although the engineering and materials analysis could be improved, the data presents and comprehensive set of potential applications of the workflow, spanning several types of cells and several types of devices. The biological data is generally convincing and of good quality. The presented procedure might therefore have some impact for the creation of prototypes in research lab, even if it boils down to other groups simply applying the same resin. 

Below I have offered a few additional comments and questions:

Comment 1: A more in depth analysis of resins content would potentially enable a discussion of why certain resins apparent work better with PDMS molding

Comment 2: The manuscript several times states "high-resolution" about the procedure, apart from being highly qualitative statement, I would also argue that in the context of soft lithography, that the resolution is not high but poor. To settle such matter, why not simply write that smallest possibly feature sizes (in the order of 100um seems like a fair statement, so "sub-millimeter")

Comment 3: The evaluation of PDMS curing seems highly qualitative. I would suggest confirming full crosslinking by extended swelling in a good solvent (Hexane perhaps) and evaluate mass of oligomers or monomers that have leached from part. 

Comment 4: rather than testing biocompatibility be seeding of cells a more convincing argument that the PDMS parts do not leach monomers or resin components would be to subject to heated water and apply analystical chemistry methods e.g. mass spec to verify that such components are not found in the media. See e.g. sup info here: https://doi.org/10.1002/advs.202001150

Comment 5: Introduction is very qualitative. I would suggest getting more specific and quantitative on e.g. spatial resolution and concept, in last paragraph of introduction 

Comment 6: Airbrush technique does not seem to be applicable for all types of designs; I suggest you elaborate and/or document limitations. 

Comment 7: It is argued several times that by getting rid of clean-room produced wafer molds the workflow is compatible with most any lab. However, however in addition to handling PDMS resins, which are not benign, the SLA workflow introduces several flammable, light sensitive, and toxic chemicals. Meaning at least a full chemistry setup with hood and chemical waste handling. Along these lines, I would argue that the main time consuming step in in traditional soft lithography is not producing the wafer, since the wafers are generally reused multiple times. To me, the main time consuming and impractical steps, relates to PDMS mixing, degassing, curing, demolding, cutting, bonding etc, which is still needed for the presented procedure. But perhaps that's the eye of the beholder. 

Comment 8: The limitations/opportunities for reusing SLA templates several times, could be investigated.

Comment 9: In the last page of section 2.2 you mention "volumetric" printing. Unless I am mistaken only traditional layer by layer SLA printing was performed (?). 

Comment 10: Figures. Figure 1C: Numbers on optical profiling data extremely hard to read, 1F: Couldn't find the arrows mentioned in caption,

---

## [Editor Report · Decision Letter 2]

8 Nov 2023

Dear Andrea,

Thank you for your patience while we considered your revised manuscript "SOL3D: Soft-lithography on 3D vat polymerised moulds for fast, versatile, and accessible microfabrication of customised multiscale cell culture devices with complex designs" for publication as a Methods and Resources at PLOS Biology. This revised version of your manuscript has been evaluated by the PLOS Biology editors, the Academic Editor.

Based on our Academic Editor's assessment of your revision, we are likely to accept this manuscript for publication, provided you satisfactorily address the following data and other policy-related requests.

* We would like to suggest a different title to improve readability:

Low-cost, versatile and highly reproducible microfabrication pipeline to generate 3D-printed customized cell culture devices with complex designs

DATA POLICY:

Regardless of the method selected, please ensure that you provide the individual numerical values that underlie the summary data displayed in the following figure panels as they are essential for readers to assess your analysis and to reproduce it: 3G, 3H, 6C, S11A, S11B, S11C

CODE POLICY

Per journal policy, as the code that you have generated is important to support the conclusions of your manuscript, we require that you make it available without restrictions upon publication. Please ensure that the code is sufficiently well documented and reusable, and that your Data Statement in the Editorial Manager submission system accurately describes where your code can be found.

* Please note that per journal policy, we do not allow the mention of "data not shown", "personal communication", "manuscript in preparation" or other references to data that is not publicly available or contained within this manuscript. Please either remove mention of these data or provide figures presenting the results and the data underlying the figure(s).

We expect to receive your revised manuscript within two weeks. 

*Published Peer Review History*

*Press*

Sincerely,

Christian

Christian Schnell, PhD

Senior Editor,

cschnell@plos.org,

PLOS Biology

---

## [Editor Report · Decision Letter 3]

5 Dec 2023

Dear Andrea,

Thank you for your patience while we considered your revised manuscript "Low-cost, versatile, and highly reproducible microfabrication pipeline to generate 3D-printed customized cell culture devices with complex designs" for publication as a Methods and Resources at PLOS Biology. 

Thank you also for addressing the editorial requests. There is only point left, which is the occurrence of "data not shown" in the third paragraph of the discussion. Per journal policy, we do not allow the mention of "data not shown", "personal communication", "manuscript in preparation" or other references to data that is not publicly available or contained within this manuscript. Please either remove mention of these data or provide figures presenting the results and the data underlying the figure(s).

We expect to receive your revised manuscript within one week. 

*Published Peer Review History*

*Press*

Sincerely,

Christian

Christian Schnell, PhD

Senior Editor,

cschnell@plos.org,

PLOS Biology

---

## [Editor Report · Decision Letter 4]

17 Jan 2024

Dear Andrea,

Thank you for the submission of your revised Methods and Resources "Low-cost, versatile, and highly reproducible microfabrication pipeline to generate 3D-printed customized cell culture devices with complex designs" for publication in PLOS Biology. On behalf of my colleagues and the Academic Editor, Chaitan Khosla, I am pleased to say that we can in principle accept your manuscript for publication, provided you address any remaining formatting and reporting issues. These will be detailed in an email you should receive within 2-3 business days from our colleagues in the journal operations team; no action is required from you until then. Please note that we will not be able to formally accept your manuscript and schedule it for publication until you have completed any requested changes.

PRESS

Sincerely, 

Christian

Christian Schnell, PhD, PhD

Senior Editor

PLOS Biology

cschnell@plos.org